# PhySwin: An Efficient and Physically-Informed Foundation Model for Multispectral Earth Observation

**Chong Tang**
University of Southampton
UCL AI Centre
Southampton, United Kingdom
chong.tang@soton.ac.uk

**Joseph Powell**
University of Manchester
Manchester, United Kingdom
joseph.powell@manchester.ac.uk

**Dirk Koch**
University of Manchester
Manchester, United Kingdom
dirk.koch@manchester.ac.uk

**Robert Mullins**
University of Cambridge
Cambridge, United Kingdom
robert.mullins@cl.cam.ac.uk

**Alex Weddell**
University of Southampton
Southampton, United Kingdom
asw@ecs.soton.ac.uk

**Jagmohan Chauhan**
University College London
UCL AI Centre
London, United Kingdom
jagmohan.chauhan@ucl.ac.uk

## Abstract

Recent progress on Remote Sensing Foundation Models (RSFMs) aims toward universal representations for Earth observation imagery. However, current efforts often scale up in size significantly without addressing efficiency constraints critical for real-world applications (e.g., onboard processing, rapid disaster response) or treat multispectral (MS) data as generic imagery, overlooking valuable physical priors. We introduce PhySwin, a foundation model for MS data that integrates physical priors with computational efficiency. PhySwin combines three innovations: (i) physics-informed pretraining objectives leveraging radiometric constraints to enhance feature learning; (ii) an efficient MixMAE formulation tailored to SwinV2 for low-FLOP, scalable pretraining; and (iii) token-efficient spectral embedding to retain spectral detail without increasing token counts. Pretrained on over 1M Sentinel-2 tiles, PhySwin achieves SOTA results (+1.32% mIoU segmentation, +0.80% F1 change detection) while reducing inference latency by up to $14.4\times$ and computational complexity by up to $43.6\times$ compared to ViT-based RSFMs.

## 1 Introduction

Earth-observation (EO) programmes now deliver petabyte-scale streams of multispectral (MS) imagery at global coverage and daily revisit rates (e.g., Sentinel-2 (S2), Landsat-8) [Roy et al., 2014, Drusch et al., 2012]. Such data supports applications ranging from precision agriculture and biodiversity monitoring to rapid flood and wildfire assessment [Gorelick et al., 2017, Van Etten et al., 2018]. Effectively utilizing the data requires models that produce general representations, enable rapid task adaptation (fine-tuning) and ensure computational efficiency for deployment.

39th Conference on Neural Information Processing Systems (NeurIPS 2025).

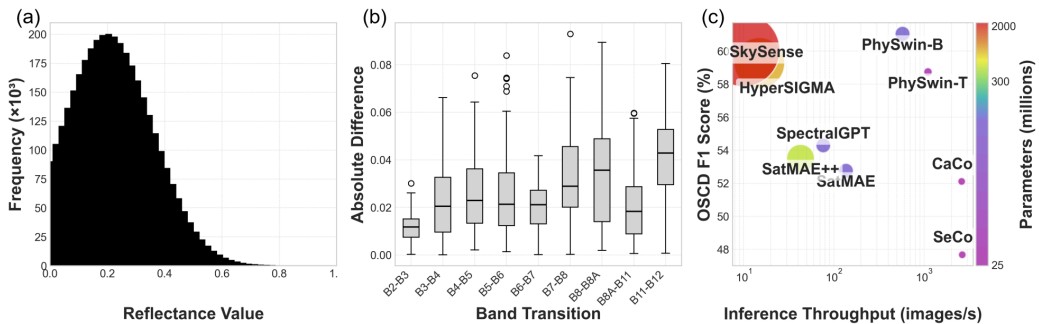

Figure 1: Motivations for PhySwin. (**a**) Bounded reflectance distribution in Sentinel-2 multispectral data, illustrating the energy-conservation property that motivates our energy-bound loss. (**b**) Smooth transitions across adjacent spectral bands, motivating the spectral-smoothness constraint used in physics-informed pretraining. (**c**) Accuracy–efficiency trade-off in change detection, highlighting PhySwin's balanced performance among EO foundation models.

Following the scaling trends in vision and NLP, Remote Sensing Foundation Models (RSFMs) have evolved from ResNet-based contrastive learners [Manas et al., 2021, Mall et al., 2023] to Vision Transformer (ViT) backbones with masked autoencoding (MAE) [Hong et al., 2024, Cong et al., 2022, Reed et al., 2023]. Subsequent models further increase capacity by fusing modalities or scaling to billion-parameter encoders [Guo et al., 2024, Wang et al., 2025a], improving accuracy but sharply raising computational costs. This has driven interest in efficiency-oriented backbones. State-space models (SSMs) adapted for EO [Wang et al., 2025b, Chen et al., 2024] offer an efficient parametric alternative, while hierarchical ViTs like Swin Transformer [Liu et al., 2021, 2022a] reduce inference complexity. Other designs address redundancy, such as HyperSIGMA's Sparse Sampling Attention [Wang et al., 2025a]. However, we argue that architectural optimizations alone cannot fully meet the dual goals of high accuracy and deployment-level efficiency required by EO systems.

Beyond computational efficiency, RSFMs often overlook the rich physics in MS data but treat them as generic image channels. Each MS band captures physically meaningful surface reflectance shaped by radiative-transfer processes [van Trigt, 1990, Hapke, 1981, Tominaga and Wandell, 1990], offering prior knowledge beyond image statistics. As shown in Fig. 1a and b, Sentinel-2 reflectance values and band transitions exhibit bounded and smooth trends, which can guide model training and promote more informative feature learning. So far, physics-aware learning has improved EO applications, for example, in crop-nitrogen retrieval [Dehghan-Shoar et al., 2024] and solar-irradiance forecasting [Liu et al., 2022b], outperforming data-driven baselines. Such priors promote faster convergence, better generalization and more informative representations. Despite this potential, most current FMs ignore these physical priors, prioritizing computational or generic vision approaches. Thus, a critical gap remains: developing models that fuse physics-awareness with computational efficiency for powerful and practical EO deployment under operational constraints.

We introduce PhySwin, a foundation model designed for MS imagery that integrates physical priors with computational efficiency through three complementary innovations: **First**, novel pretraining objectives embed radiometric constraints, including adjacent-band spectral smoothness and energy conservation, into self-supervised learning (SSL). This leads to more robust and meaningful feature representations. **Second**, PhySwin refines the Mixed and Masked Autoencoding (MixMAE) method [Liu et al., 2023] to address the limitations of standard MAE and computationally intensive MIM variants in hierarchical models. Our approach leverages SwinV2's shifted window mechanism for scalable, low-cost pretraining while enabling effective cross-window interaction for global context modeling. **Third**, PhySwin embeds MS data efficiently by grouping spectral bands and concatenating distinct feature subspaces per patch to retain spectral detail without increasing token counts. During pretraining, spectral group masking randomly removes entire groups of features to simulate spectral variability and enhance robustness. Pretrained on over one million S2 tiles and evaluated on six EO benchmarks, PhySwin outperforms strong ViT baselines, improving segmentation mIoU by +1.32% and change detection F1 by +0.80%, while reducing inference latency and computational cost by up to 14.4× and 43.6×, respectively. As shown in Fig. 1c, PhySwin achieves a favorable balance compared against other state-of-the-art (SOTA) baselines. Our main technical contributions are:

- **The first integration of physics-aware objectives** into large-scale MS foundation model pretraining, using physical constraints to generate high-fidelity features that overcome the typical accuracy trade-offs associated with computationally efficient architectures.

- **A novel MixMAE formulation for SwinV2** that enables highly efficient and resolution-flexible pretraining on large EO datasets.

- **A token-efficient spectral embedding technique** that efficiently represents comprehensive MS band information without increasing token counts, addressing a common efficiency issue in existing RSFMs.

## 2 Related Work

**Backbone Evolution and Pretraining Paradigms.** Early EO FMs adapted contrastive learning on CNNs to leverage satellite time-series consistency [Manas et al., 2021, Mall et al., 2023]. Subsequent work adopted Transformer backbones, particularly as MAE techniques proved effective for large-scale pretraining [Cong et al., 2022]. This shift stimulated architectural diversification from standard ViTs compatible with basic MAE to hierarchical Swin-style transformers [Guo et al., 2024] requiring adapted pretraining, and to efficiency-focused state-space models [Chen et al., 2024, Wang et al., 2025b]. Additional innovations include MAE variants tailored for scale [Noman et al., 2024, Reed et al., 2023] or spectral data [Hong et al., 2024], multi-tasking [Wang et al., 2024], continual learning [Mendieta et al., 2023] and multi-modal fusion [Guo et al., 2024, Nedungadi et al., 2024]. While significant progress has been driven by scaling models and datasets to enhance performance, recent research has increasingly prioritized computational efficiency and scalability for practical deployment, particularly on resource-constrained platforms [Chen et al., 2025, Wang et al., 2025b].

**Efficient Foundation Models.** FM architectural choices strongly influence computational efficiency. Hierarchical transformers like Swin [Liu et al., 2022a] reduce complexity from quadratic to linear compared to standard ViTs through windowed attention. SSMs [Gu and Dao, 2023] offer linear scaling for long sequences, though applicability may depend on task-specific adaptations or pretraining strategies. These designs often complicate pretraining. For example, MAE's patch discarding conflicts with the structure of hierarchical models [Liu et al., 2023]. A common alternative uses MIM with `MASK` tokens (e.g., SimMIM [Xie et al., 2022]), though these non-informative tokens add inefficiency and pretrain-finetune discrepancies. Other approaches include supervised pretraining [Bastani et al., 2023] and contrastive learning, but MAE's reconstruction objective often produces richer features for dense prediction tasks [He et al., 2022] while avoiding the label demands of supervised methods. Returning to our focus on EO tasks, the existing work on FM architectures and pretraining methods inspired our realization that designing efficient RSFMs requires balancing architectural benefits, pretraining complexity, and performance trade-offs.

**Physics-informed ML.** Incorporating domain knowledge enhances foundation models. Physics-informed learning, which integrates physical laws as priors, has shown promise in climate modeling, materials science, and manufacturing [Karniadakis et al., 2021, Dehghan-Shoar et al., 2024]. Applying such priors in large-scale RSFM pretraining remains underexplored. MS imagery is well-suited due to physical principles governing surface reflectance: smooth spectral variation across adjacent bands [van Trigt, 1990, Tominaga and Wandell, 1990] and energy conservation bounding reflectance between 0 and 1 [Hapke, 1981]. Leveraging these priors during pretraining improves representational power and efficiency. PhySwin follows this principle, combining physics-informed objectives with efficient architectures for high accuracy and practical deployment.

## 3 PhySwin

We identify three core challenges in building efficient RSFMs: (i) pretraining inefficiencies of hierarchical architectures; (ii) the explosion of token counts when naively handling MS data; and (iii) the degradation of representation quality under tight compute budgets. To address these challenges, we propose PhySwin, an efficiency-oriented RSFM that leverages refined MixMAE pretraining, token-efficient embedding and physics-aware objectives to achieve high performance on EO tasks with significantly reduced computational cost (Fig. 2), detailed in the following subsections.

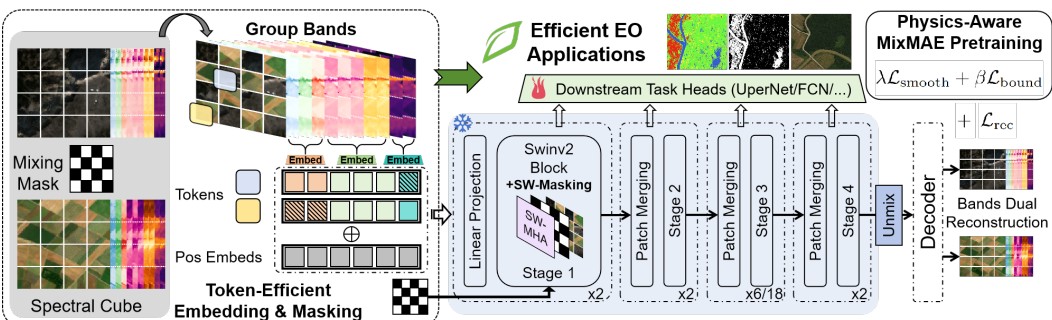

Figure 2: PhySwin framework: Two MS cubes are mixed via spatial masking, then grouped by band types and embedded. The SwinV2 encoder with SW-Masking processes the inputs through hierarchical stages. Pretraining is guided by $\mathcal{L}_{\text{rec}}$ and regularization: $\mathcal{L}_{\text{smooth}}$ and $\mathcal{L}_{\text{bound}}$. PhySwin achieves high performance across diverse downstream tasks with improved efficiency.

## 3.1 Physically-Informed Pretraining Objective

Multispectral sensors (e.g., S2, Landsat-8) measure surface reflectance across $B$ bands, each encoding distinct material properties (e.g., vegetation health in Near-Infrared (NIR), moisture in Short-Wave Infrared (SWIR) spectroscopy). We embed two radiometric priors as regularizers on the reconstructed reflectance vector $\hat{\mathbf{r}} = [\hat{r}_1, \ldots, \hat{r}_B] \in \mathbb{R}^B$:

**Spectral Smoothness.** Natural surface reflectance spectra are usually a smooth function of wavelength, lacking sharp or random fluctuations between adjacent bands. This well-established property stems from the continuous nature of light interactions with materials and is supported by both theoretical analysis [van Trigt, 1990] and empirical studies [Tominaga and Wandell, 1990] (as shown in Fig. 1b). To discourage fitting noise or sensor artifacts, we impose a smoothness regularization based on the first-order finite difference:

$$\mathcal{L}_{\text{smooth}} = \sum_{b=1}^{B-1} \left( \hat{r}_{b+1} - \hat{r}_b \right)^2, \tag{1}$$

which penalizes high-frequency spectral fluctuations.

**Energy Conservation.** Physically, surface reflectance quantifies the fraction of incident electromagnetic energy reflected at each wavelength. Due to the energy conservation, this value is inherently bounded, typically within the range $[0, 1]$ [Hapke, 1981] (seen in Fig. 1a). Incorporating this constraint guides the model to operate within a physically realistic output space. To enforce this constraint, we apply:

$$\mathcal{L}_{\text{bound}} = \sum_{b=1}^{B} \left[ \text{ReLU}(-\hat{r}_b) + \text{ReLU}(\hat{r}_b - 1.2) \right]. \tag{2}$$

Both terms remain fully differentiable. For implementation, we relax the upper bound to $1.2$ to accommodate sensor noise.

## 3.2 Efficient Pretraining via Refined MixMAE on SwinV2

PhySwin adopts Swin Transformer V2 [Liu et al., 2022a] for its computational efficiency: the Swin family scales linearly with input size via the shifted window self-attention (SW-MSA). Swin V2 extends V1 with improved scaling and a log-spaced continuous position bias [Liu et al., 2021], which enhances transferability across resolutions and window sizes and is particularly useful for the diverse spatial scales of EO.

As discussed in Section 2, efficiently pretraining hierarchical ViTs remains nontrivial. To address this bottleneck, MixMAE [Liu et al., 2023] is proposed. Its core idea is to replace masked patches from one image ($x_1^p$) with visible patches from a second image ($x_2^p$) using a random binary mask $M$. The resulting mixed input contains only real image tokens (seen the mixing process in Fig. 2):

$$\hat{x}_m^p = x_1^p \odot M + x_2^p \odot (1 - M).$$

A lightweight decoder reconstructs both original images under a dual reconstruction loss:

$$\mathcal{L}_{\text{rec}} = \|(y_1^p - x_1^p) \odot (1 - M)\|_2^2 + \|(y_2^p - x_2^p) \odot M\|_2^2, \tag{3}$$

where $y_1^p$ and $y_2^p$ are the reconstructions of $x_1^p$ and $x_2^p$, respectively. This dual reconstruction objective ensures that each source image is reconstructed solely from its own unmasked context, encouraging the model to infer missing regions using within-source evidence rather than cross-source tokens. During encoding, tokens attend only to others from the same source image, as dictated by $M$, preventing information leakage. This design enables MAE-style pretraining to be applied to hierarchical ViTs while avoiding non-informative `[MASK]` tokens in the encoder. Notably, the original MixMAE implementation disables SW-MSA and instead relies on large fixed windows to model global context.

PhySwin refines MixMAE for SwinV2 by retaining the SW-MSA mechanism to support improved cross-window interactions and global context modeling (seen in SwinV2 block in Fig. 2). Specifically, we introduce SW-Masking, in which the binary mask $M$ is spatially shifted in alignment with SW-MSA configurations. We argue that coordinating the masking pattern with shifted windows promotes more robust feature learning across window boundaries, preserves spatial continuity, and reduces reconstruction artifacts compared to fixed-window schemes.

### 3.3 Token-Efficient Grouped Spectral Embedding and Masking

Standard ViTs embed non-overlapped RGB patches ($\mathbb{R}^{H \times W \times 3}$) into $D$-dimensional tokens, producing $N$ tokens for $N$ patches. Directly extending this to MS data ($\mathbb{R}^{H \times W \times B}$, with $B \gg 3$) may diminish critical band-specific information. Prior works address this via methods like grouped channel embeddings [Cong et al., 2022], 3D spatial-spectral tokens [Hong et al., 2024] or separate feature branches [Wang et al., 2025a]. However, these techniques increase token counts and may overlook correlations between spectral bands.

We propose a token-efficient embedding strategy that preserves the token count comparable to standard processing while retaining the rich spectral structure of MS data. Given a sample $x \in \mathbb{R}^{H \times W \times B}$, we partition the $B$ spectral bands into $G$ physically coherent groups (e.g., visible, NIR, SWIR ), $x^{(g)} \in \mathbb{R}^{H \times W \times B_g}$ with $\sum_{g=1}^{G} D_g = D$. For each spatial position $(i, j)$, we extract group-wise local patches $x_{i,j}^{(g)} \in \mathbb{R}^{P \times P \times B_g}$. Each group $g$ is then processed by a lightweight, group-specific embedding function $f_g : \mathbb{R}^{P \times P \times B_g} \to \mathbb{R}^{D_g}$, and the final token $e_{i,j} \in \mathbb{R}^D$ is constructed by concatenating the outputs across all groups (Fig. 2):

$$e_{i,j} = \text{Concat}\left(f_1(x_{i,j}^{(1)}), f_2(x_{i,j}^{(2)}), \dots, f_G(x_{i,j}^{(G)})\right) \in \mathbb{R}^D. \tag{4}$$

This yields a single embedding vector $e_{i,j} \in \mathbb{R}^D$ per spatial location. Therefore, an input image divided into $N$ spatial patches produces exactly $N$ tokens, maintaining the sequence length efficiency.

Building on this design, we further introduce *Spectral Group Masking* (MaskSpec) during pretraining to improve representation robustness and simulate real-world spectral variations. PhySwin randomly zeros out one or more spectral group subspaces $f_g(x_{i,j}^{(g)})$ within each token $e_{i,j}$, encouraging representations that are less reliant on any single spectral group. This spectral masking complements the spatial masking of our refined MixMAE framework (Section 3.2), where we use a fixed 50% mixing ratio. By operating in the spectral domain, we can modulate the overall pretraining difficulty. To summarize the combined masking strategy, for each token $e_{i,j}$, we apply:

$$\tilde{e}_{i,j} = \text{MixMAE}\left(\text{MaskSpec}(e_{i,j}^{(1)}, M_g^{(1)}), \ \text{MaskSpec}(e_{i,j}^{(2)}, M_g^{(2)}), \ M\right), \tag{5}$$

where $M_g^{(*)} \in \{0,1\}^G$ are independent group masks. This joint masking regulates spatial and spectral exposure, yielding more generalizable features.

### 3.4 Pretraining Details

**Datasets and Preprocessing.** Following a two-stage pretraining strategy similar to Spectral-GPT [Hong et al., 2024], PhySwin is first trained for 200 epochs on FMoW-S2 (about $712,000$ samples) [Christie et al., 2018] with $96 \times 96$ S2 tiles, then for 100 epochs on BigEarthNet-S2 (about $590,000$ samples) [Sumbul et al., 2019] with $128 \times 128$ tiles. Raw reflectance values are normalized

by $1/10,000$ to approximate the $[0, 1]$ range. We exclude three 60-meter resolution S2 bands (B01, B09, B10) following SpectralGPT and SatMAE [Hong et al., 2024, Cong et al., 2022]. The retained 10 bands are grouped into: Visible (B02, B03, B04), Red-Edge/NIR (B05, B06, B07, B08, B8A), and SWIR (B11, B12) corresponding to our embedding method (Section 3.3).

**Backbone Configurations.** PhySwin is developed using two Swin Transformer V2 variants, both configured with a base input resolution of $128 \times 128$, a patch size of 4 and a window size of 7.

| Variant | Embedding Dim | Depths | Num Heads | #Params (M) |
|---|---|---|---|---|
| PhySwin-Tiny (T) | 96 | [2, 2, 6, 2] | [3, 6, 12, 24] | 29 |
| PhySwin-Base (B) | 128 | [2, 2, 18, 2] | [4, 8, 16, 32] | 88 |

**Training Setup.** PhySwin was pretrained on eight NVIDIA RTX 8000 GPUs using mixed precision. We employed the AdamW optimizer with a cosine learning rate schedule and a 15-epoch linear warmup. The training objective

$$\mathcal{L}_{\text{total}} = \mathcal{L}_{\text{rec}} + \lambda \mathcal{L}_{\text{smooth}} + \beta \mathcal{L}_{\text{bound}}. \tag{6}$$

Weights were fixed during pretraining at $\lambda = 0.25$, $\beta = 0.1$. Full hyperparameters are detailed in Appendix A.

# 4 Experiment

In this section, we evaluate the effectiveness and efficiency of PhySwin against SOTA RSFMs across four downstream EO tasks. To ensure reproducibility, PhySwin is built on the Hugging Face Transformers library [Wolf et al., 2020], following its coding standards and API conventions. PhySwin models and SOTA baselines replicated in this study are fine-tuned under a unified protocol, including SeCo (ResNet50), CACo (ResNet50), SatMAE (ViT-B), SatMAE++ (ViT-L) and SpectralGPT (ViT-B). Performance for other baselines (e.g., SkySense, HyperSIGMA) is reported from the literature. Across all tasks, PhySwin uses the following S2 bands: native (B02, B03, B04, B08) and resampled to 10m (B05, B06, B07, B8A, B11, B12). *Extended experimental results, including additional benchmarks and ablation studies, are provided in Appendix 5.*

## 4.1 Benchmarks

**Semantic Segmentation.** Performance is evaluated on two benchmarks. SegMunich [Hong et al., 2024] contains overlapping $128 \times 128$ pixel tiles (50% overlap) with 13 Land Use and Land Cover (LULC) classes. DynamicEarthNet-Sentinel2 (Dyna.-S2) [Toker et al., 2022] consists of monthly S2 composites from January 2018 to December 2019, tiled into $256 \times 256$ patches. Mean Intersection over Union (mIoU) is reported for both.

**Change Detection.** Evaluated on OSCD [Daudt et al., 2018], comprising 24 S2 image pairs (14 train, 10 test), split into non-overlapping $96 \times 96$ patches as the SkySense protocol [Guo et al., 2024]. Precision, Recall, and F1 are reported. Dyna.-S2 image pairs are formed from monthly composites, tiled into $96 \times 96$ patches, and semantic change segmentation score (SCS) [Toker et al., 2022] is calculated from 7-class segmentation labels.

**Scene and Multi-Label Land Cover Classification.** Scene classification is evaluated on two benchmarks. FMoW-S2 contains 62 land-use types; we follow the SatMAE splits [Cong et al., 2022] and report top-1 accuracy. EuroSAT [Helber et al., 2018, 2019] comprises $27,000$ S2 images ($64 \times 64$ pixels) across 10 land cover classes; we follow [Helber et al., 2019] and report overall accuracy (OA). Multi-label classification is evaluated on BigEarthNet, which contains $120 \times 120$ S2 images from 10 countries annotated with 19 land cover classes, using official splits [Clasen et al., 2024]. We report mean average precision (mAP).

Table 1: Semantic segmentation performance.[1]

| Model | SegMunich | ΔmIoU (%) | Dyna.-S2 | ΔmIoU (%) |
|---|---|---|---|---|
| SeCo | 45.92 | −5.08 | 40.19 | −6.01 |
| CACo | 44.87 | −6.13 | 41.50 | −4.70 |
| SatMAE | 48.71 | −2.29 | 38.73 | −7.47 |
| SatMAE++ | 50.62 | −0.38 | 42.87 | −3.33 |
| SpectralGPT | **51.00** | - | 44.72 | −1.48 |
| SkySense[†] | – | – | **46.20** | - |
| PhySwin-T | 49.53 | −1.47 | 43.80 | −2.40 |
| PhySwin-B | **52.32** | +1.32 | **46.53** | +0.33 |

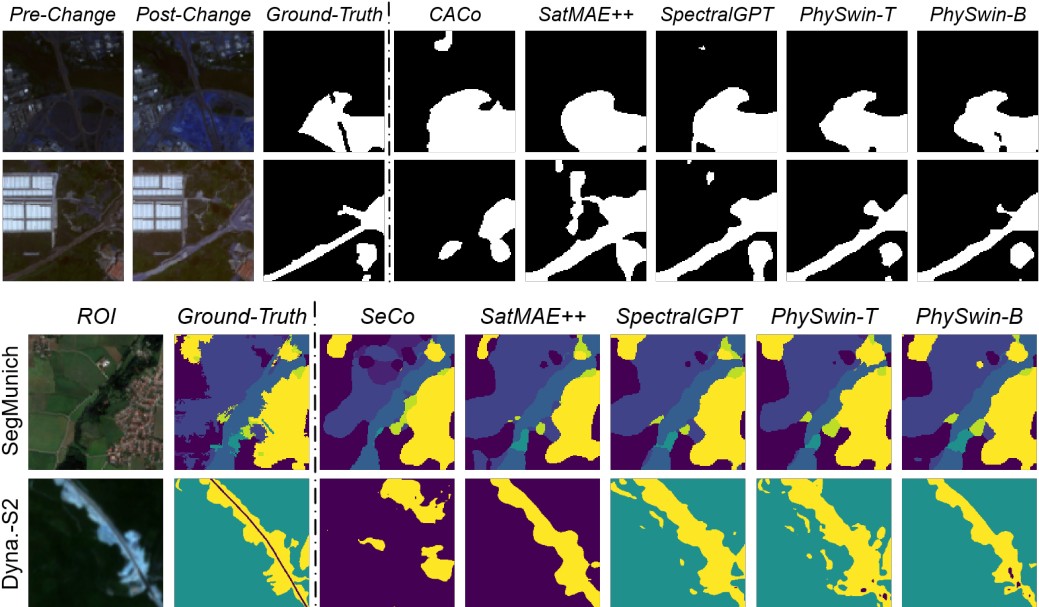

Figure 3: Performance visualization of downstream tasks. TOP: OSCD change detection. BOTTOM: SegMunich and Dyna.S2 semantic segmentation. Colors follow each dataset's official palette.

## 4.2 Downstream Tasks Performance

**Semantic Segmentation Results.** Table 1 reports mIoU on SegMunich and Dyna.-S2 compared to RSFM baselines. On SegMunich, PhySwin-T achieves 49.53%, ranking third behind SpectralGPT (51.00%) and SatMAE++ (50.62%) and surpassing SatMAE ViT-B (48.71%). On Dyna.-S2, it reaches 43.80%, outperforming both SatMAE variants while using only one-third the tokens of SpectralGPT. PhySwin-B sets new SOTA results with 52.32% (+1.32% over SpectralGPT) on SegMunich and 46.53% (+0.33% over SkySense) on Dyna.-S2. These results show PhySwin improves dense prediction accuracy without increasing model size or runtime. Qualitatively (Fig. 3), PhySwin produces sharper boundaries and better recovers thin structures, such as narrow roads and small bodies, matching the quantitative improvements in Table 1.

**Change Detection Results.** Table 2 reports F1, precision, and recall on OSCD and SCS on Dyna.-S2. PhySwin-T achieves 58.74% F1, outperforming ChangeMamba (57.20%) and all ViT-based methods, and records 17.61% SCS, closely matching SkySense (18.00%). PhySwin-B reaches 61.05% F1 (+0.80% over DynamicVis) and 18.13% SCS (+0.13% over SkySense). DynamicVis achieves the highest precision (79.41%) but low recall (48.36%), while PhySwin-B offers a better balance (67.97% precision, 55.41% recall). These gains, also shown in Fig. 3, demonstrate that our physics-informed

---

[1]The top two results for each dataset are highlighted in **bold**. Δ denotes the gain over the best published RSFM baseline. [†] indicates results reported from the corresponding literature.

Table 2: Change detection performance.

| Model | OSCD | | | ΔF1% | Dyna.-S2 | ΔSCS% |
| | F1% | P% | R% | | SCS% | |
|---|---|---|---|---|---|---|
| SkySense[†] | 60.06 | – | – | -0.19 | **18.00** | – |
| HyperSIGMA[†] | 59.28 | 59.12 | **59.45** | -0.97 | – | – |
| SpectralGPT | 54.29 | 52.39 | 57.20 | -5.96 | 17.43 | -0.57 |
| ChangeMamba[†] | 57.20 | 56.08 | **58.36** | -3.05 | – | – |
| DynamicVis[†] | **60.25** | **79.41** | 48.36 | – | – | – |
| SpatSIGMA[†] | 58.53 | 64.59 | 53.50 | -1.72 | – | – |
| SeCo | 47.67 | 63.21 | 38.26 | -12.58 | 16.00 | -2.00 |
| CACo | 52.11 | 62.87 | 44.49 | -8.14 | 16.53 | -1.47 |
| SatMAE | 52.76 | 55.18 | 50.54 | -7.49 | 16.20 | -1.80 |
| SatMAE++ | 55.31 | 58.07 | 52.80 | -4.94 | 17.88 | -0.12 |
| PhySwin-T | 58.74 | 65.93 | 52.96 | -1.51 | 17.61 | -0.39 |
| PhySwin-B | **61.05** | **67.97** | 55.41 | +0.80 | **18.13** | +0.13 |

Table 3: Multi-label and scene classification performance.

| Model | BigEarthNet | | FMoW-S2 | | EuroSAT | |
| | mAP% | ΔmAP% | Top-1 Acc% | ΔAcc% | OA% | ΔOA% |
|---|---|---|---|---|---|---|
| SkySense[†] | **92.09** | 0.00 | **64.38** | -0.86 | — | — |
| SatMAE | 82.13 | -9.96 | 63.84 | -1.40 | 98.98 | -0.23 |
| SatMAE++ | 85.11 | -6.98 | **65.24** | 0.00 | **99.04** | -0.17 |
| SpectralGPT | **88.22** | -3.87 | 64.21 | -1.03 | **99.21** | 0.00 |
| SeCo | 87.81 | -4.28 | 51.65 | -13.59 | 95.63 | -3.58 |
| CACo | 87.00 | -5.09 | 50.72 | -14.52 | 95.90 | -3.31 |
| PhySwin-T | 86.63 | -5.46 | 59.26 | -5.98 | 97.25 | -1.96 |
| PhySwin-B | 87.93 | -4.16 | 63.11 | -2.13 | 98.73 | -0.48 |

pretraining reduces false positives and better captures subtle spectral changes, producing cleaner masks and more precise localization of small variations.

**Multi-label and Scene Classification.** Table 3 shows PhySwin achieves competitive, near-SOTA results across all three benchmarks. On BigEarthNet, PhySwin-B reaches 87.93% mAP (third, within 4.16% of SkySense), while PhySwin-T records 86.63%, outperforming SeCo and CACo. On FMoW-S2, PhySwin-B achieves 63.11% top-1 accuracy (2.13% below SatMAE++), and PhySwin-T scores 59.26%. On EuroSAT, PhySwin-B attains 98.73% overall accuracy (0.48% below SOTA SpectralGPT), and PhySwin-T posts 97.25%. These small gaps likely reflect Swin's windowed attention emphasizing local context. We consider this an acceptable trade-off given PhySwin's faster inference and reduced memory compared to ViT-based RSFMs (discussed next).

## 4.3 Efficiency Analysis

Table 4 compares feature-extraction efficiency at $128 \times 128$ input on an RTX 2000 Ada GPU. CNNs like SeCo offer high throughput with low compute and memory cost but lower accuracy. ViTs improve accuracy but lose efficiency when scaled (e.g., SatMAE++ vs. SatMAE) or with more tokens (e.g., SpectralGPT vs. SatMAE), shown by higher FLOPs, memory and lower throughput. PhySwin balances these trade-offs: PhySwin-T delivers $14.4\times$ higher throughput than SpectralGPT with $43.6\times$ fewer FLOPs and 62.5% less memory. PhySwin-B exceeds SatMAE++ with $13.5\times$ speedup and 64% memory reduction. These results show PhySwin improves efficiency over ViTs, approaching CNN-level efficiency without sacrificing accuracy advantages (as shown in Fig. 1c).

## 4.4 Ablation Study

We perform ablations on PhySwin-B to isolate the contributions of each design choice. All experiments use the same evaluation heads and data splits as in Sec. 4. **Physically-Informed Losses Ablation.** We ablate physics-informed losses on PhySwin-B (Table 5a). The baseline without physics priors shows the lowest scores. Adding spectral smoothness ($L_{\text{smooth}}$) or reflectance bounding ($L_{\text{bound}}$)

Table 4: Inference efficiency (feature extraction) at $128 \times 128$ input, batch size 4, after 10 warm-up iterations on a single RTX 2000 Ada GPU. Type = model; Params = parameters; Throughput = images/s; Mem = peak GPU memory. CACo is excluded as it shares SeCo's ResNet50 backbone.

| Model | Type | Params (M) | FLOPs (G) | Throughput (imgs/s) | Mem (GB) |
|---|---|---|---|---|---|
| SpectralGPT | ViT-B | 85.4 | 87.2 | 76.4 | 0.8 |
| SatMAE | ViT-B | 85.7 | 65.5 | 136.4 | 0.5 |
| SatMAE++ | ViT-L | 303.2 | 232.6 | 42.8 | 1.4 |
| SeCo | ResNet-50 | 23.5 | 1.4 | **2642.8** | 0.1 |
| PhySwin-T | SwinV2-T | 28.5 | 2.0 | **1102.0** | 0.3 |
| PhySwin-B | SwinV2-B | 87.9 | 7.6 | **576.8** | 0.5 |

Table 5: Ablation studies on physics-informed losses and masking schemes. **Seg**: SegMunich (mIoU %), **CD**: OSCD (F1 %), **CLS**: FMoW-S2 (Top-1 %), **Throughput** in images per second (img/s).

| Variant | Seg (%) | CD (%) | CLS (%) |
|---|---|---|---|
| No physics | 47.73 | 53.37 | 57.66 |
| Lsmooth only | **50.91** | **58.99** | **61.93** |
| Lbound only | 49.78 | 56.42 | 59.46 |
| Lsmooth + Lbound | **52.32** | **61.05** | **63.11** |

| Mask type | Seg (%) | CD (%) | Throughput |
|---|---|---|---|
| SimMIM | 47.79 | 52.46 | **188.50** |
| MixMAE | 52.11 | 59.33 | 153.26 |
| Ours | **52.32** | **61.05** | 149.84 |

(a) Ablation of physics-informed losses.
(b) Ablation of masking schemes.

significantly improves performance, with $L_{\text{smooth}}$ obtaining larger gains by better capturing inter-band relationships. Combining both losses achieves the best results, confirming complementary effects. These results demonstrate that physics-informed constraints enhance feature quality.

**Pretraining Methods.** Table 5b compares pretraining methods for SwinV2-B beyond throughput. SimMIM shows highest throughput but lowest accuracy, as [MASK] tokens offer less effective feature learning. MixMAE improves accuracy via dual reconstruction and better token use. Our method enhances MixMAE with SW-MSA and SW-Masking, achieving top accuracy with minimal throughput loss. These results highlight the advantage of effective information learning (MixMAE, ours), with SW-MSA providing additional global context.

**Spectral Embedding Ablation.** Table 6 evaluates five spectral embedding strategies. Since the spectral structure is ignored, the naive flattened processing has the highest throughput but the lowest accuracy. SpectralGPT's 3D Patch improves accuracy by splitting spectra into 3-band tokens, but significantly reduces throughput. SatMAE's Grouped Embedding (GE) uses one-third fewer tokens than SpectralGPT and boosts performance. Our token-efficient GE packs spectral groups into single embeddings, matching SatMAE GE accuracy with $3\times$ higher throughput. Adding MaskSpec subspace masking further enhances robustness, delivering the best accuracy-efficiency balance.

**Hyperparameter Sensitivity.** We tested the physics-loss weights $\lambda \in \{0.1, 0.25, 0.5\}$, $\beta \in \{0.05, 0.1, 0.2\}$, and MixMAE mixing ratios of 50%, 67% and 75%. Across these settings, segmentation mIoU and change-detection F1 vary by less than 2%, demonstrating that our defaults ($\lambda = 0.25$, $\beta = 0.1$, mix=50%) lie in a stable region.

## 5   Conclusion and Limitations

We presented PhySwin, a novel foundation model for multispectral EO that integrates three key innovations: (1) *physics-informed pretraining* via spectral-smoothness and energy-conservation losses, (2) *refined MixMAE* tailored to SwinV2, and (3) *token-efficient embedding* with MaskSpec. Pretrained on over one million Sentinel-2 tiles and evaluated across various downstream benchmarks, PhySwin achieves SOTA performance while significantly reducing both computational complexity and inference latency. We observe that model size correlates positively with FM performance. However, in ViT-based models, scaling these factors drastically increases fine-tuning and inference complexity, limiting practical deployment. Our results confirm that the hierarchical structure mitigates

Table 6: Ablation of embedding strategies (PhySwin-B Inference). 3D Patch follows SpectralGPT; Grouped Embedding (GE) group bands for embedding; MaskSpec adds random subspace masking.

| Variant | Seg mIoU (%) | OSCD F1 (%) | FMoW-S2 Top-1 (%) | Throughput (img/s) |
|---|---|---|---|---|
| Naive | 43.21 | 44.76 | 52.73 | **599.38** |
| 3D Patch | **53.47** | **61.34** | **64.13** | 144.32 |
| GE (SatMAE) | **54.22** | 60.11 | 63.77 | 155.98 |
| GE (Ours) | 52.03 | 60.89 | 63.02 | 544.25 |
| GE+MaskSpec (Ours) | 52.32 | **61.05** | **63.11** | **584.58** |

these issues, offering multi-stage features beneficial for challenging EO tasks such as change detection. PhySwin further amplifies these strengths, delivering an optimal balance between computational efficiency and downstream task accuracy.

While PhySwin shows a modest performance gap on certain classification benchmarks, we attribute this mainly to SwinV2's hierarchical design, which is less suited for global pooling classification. Model scaling was not the focus of this study, though larger variants may yield further gains. Additional limitations include the spectral-smoothness prior's sensitivity to residual band misregistration and the use of single-timestamp pretraining, adopted to preserve an edge-friendly compute budget. Future work will extend PhySwin toward multi-temporal pretraining through efficient temporal patching and spatio-temporal masking, and explore misregistration-robust smoothness variants, hybrid architectures, and advanced radiative-transfer constraints (e.g., BRDF models, atmospheric correction) for multi-modal fusion such as SAR–MS.

## Acknowledgments and Disclosure of Funding

This work was supported by the Engineering and Physical Sciences Research Council (EP-SRC), United Kingdom, under the project *Perfect Recollection for Clearer Insight* (grant number EP/Y036077/1).

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

## Appendix A: Training and Implementation Details

**Compute environment.** Training was conducted on two nodes equipped with 8 NVIDIA Quadro RTX 8000 GPUs (each with 48 GB memory). Mixed-precision training (AMP) was enabled throughout. Distributed training was implemented via `torchrun` with NCCL backend, using socket-based transport for stability. Inference evaluations were performed on a single RTX 8000 GPU and also a single NVIDIA RTX 2000 Ada GPU (8 GB VRAM).

**Training configuration.** We summarize the key hyperparameters in Table 7.

Table 7: Training configurations for progressive pretraining.

| Parameter | Stage 1: FMoW-S2 | Stage 2: BigEarthNet-S2 |
|---|---|---|
| Image resolution | $96 \times 96$ | $128 \times 128$ |
| Epochs | 200 | 100 |
| Batch size per GPU | 256 | 256 |
| Total batch size | 2048 | 2048 |
| Learning rate | $1 \times 10^{-4}$ | $5 \times 10^{-5}$ |
| Weight decay | $5 \times 10^{-5}$ | $5 \times 10^{-5}$ |
| Warmup epochs | 10 | 5 |
| Scheduler | Cosine decay | Cosine decay |
| Gradient clipping | max_norm = 1.0 | max_norm = 1.0 |
| Steps per epoch | 695 | 537 |
| Warmup steps | 6950 | 2685 |

## Appendix B: Dataset Processing and Band Grouping

We apply a unified preprocessing pipeline across all Sentinel-2 datasets used in this work. Key preprocessing steps include band selection, normalization, and spatial resizing, detailed in Table 8. Bands B01, B09, and B10 are excluded due to their coarse 60 m resolution. All retained bands are grouped into physically coherent spectral categories for structured encoding.

Table 8: The summary of common preprocessing and dataset-specific properties.

| Common Preprocessing Across All Datasets | |
|---|---|
| Normalization | Pixel values divided by 10,000 |
| Valid reflectance range | Clipped to $[0, 1.2]$ |
| Resizing method | Bicubic interpolation to fixed spatial size |
| Excluded bands | B01, B09, B10 (60 m resolution) |
| Retained bands | B02–B08A, B11, B12 (10 bands) |
| Spectral groupings | Visible: B02, B03, B04 |
| | Red-Edge/NIR: B05–B08, B8A |
| | SWIR: B11, B12 |

| Dataset-Specific Details | | | |
|---|---|---|---|
| **Dataset** | **Tile Size** | **#Tiles** | **Task** |
| FMoW-S2 | $96 \times 96$ | 712,874 | Pretraining & scene-level classification |
| BigEarthNet-S2 | $128 \times 128$ | 549,488 | Pretraining & multi-label classification |
| OSCD | $96 \times 96$ | 336 pairs | Change detection |
| SegMunich | $128 \times 128$ | 8,430 | Semantic segmentation |
| DynaS2 | $256 \times 256$ | 5,472 pairs | Multi-temporal change detection |
| EuroSAT | $64 \times 64$ | 27,000 | Land cover classification |

**Dataset licenses and sources.**

- **BigEarthNet-S2:** `https://bigearth.net`. Licensed under the **Community Data License Agreement – Permissive, Version 1.0**.

- **FMoW-S2:** `https://github.com/fMoW/dataset`. Released under the **Functional Map of the World Challenge Public License**. Openly available for non-commercial research use.
- **OSCD (Onera Satellite Change Detection):** `https://rcdaudt.github.io/oscd/`. Publicly released for academic benchmarking; **no explicit license** provided.
- **DynaS2 (DynamicEarthNet Sentinel-2):** `https://mediatum.ub.tum.de/1650201`. Licensed under **Creative Commons Attribution-ShareAlike 4.0 International (CC BY-SA 4.0)**.
- **EuroSAT:** `https://github.com/phelber/eurosat`. Licensed under the **MIT License**.
- **SegMunich:** `https://huggingface.co/datasets/earthflow/SegMunich`. Distributed under **Creative Commons Attribution-NonCommercial 4.0 International (CC BY-NC 4.0)**.

## Appendix C: Expanded Ablation Studies

All ablations in this section are conducted using the PhySwin-T model pretrained on the BigEarthNet-S2 dataset for 100 epochs. So, the reported results may slightly differ from those presented in Section 4. Evaluation is performed on three downstream tasks: SegMunich (semantic segmentation), OSCD (change detection), and EuroSAT (land cover classification).

### C.1 Physics Loss Weights

Table 9: Sensitivity to physics-informed loss weights $(\lambda, \beta)$. Mixing ratio is fixed at 50%, and spectral grouping follows the default setting in Section 4. The best results are highlighted in **bold**.

| $(\lambda, \beta)$ | SegMunich mIoU (%) | OSCD F1 (%) | EuroSAT OA (%) |
|---|---|---|---|
| (0.1, 0.05) | 47.97 | 56.33 | 96.64 |
| **(0.25, 0.1)**\* | **48.46** | 56.98 | **96.88** |
| (0.5, 0.2) | 48.21 | **57.03** | 96.27 |

### C.2 Mixing Ratio Effects

Table 10: Ablation on MixMAE spatial mixing ratio. Physics-informed loss weights are fixed at $(\lambda = 0.25, \beta = 0.1)$, and spectral grouping follows the default setting in Section 4. The best results are highlighted in **bold**.

| Mixing Ratio | SegMunich mIoU (%) | OSCD F1 (%) | EuroSAT OA (%) |
|---|---|---|---|
| **50%**\* | **48.46** | **56.98** | **96.88** |
| 67% | 45.29 | 55.63 | 96.01 |
| 75% | 42.97 | 53.61 | 94.79 |

### C.3 Spectral Grouping Variants

Table 11: Performance of different spectral grouping strategies. Physics loss weights are fixed at $(\lambda = 0.25, \beta = 0.1)$, and the MixMAE spatial mixing ratio is fixed at 50%. Default grouping is: Visible (B02–B04), RedEdge+NIR (B05–B08A), SWIR (B11–B12). The best two results are highlighted in **bold**.

| Grouping Scheme | #Groups | SegMunich mIoU (%) | OSCD F1 (%) | EuroSAT OA (%) |
|---|---|---|---|---|
| **Visible \| RedEdge+NIR \| SWIR**\* | 3 | **48.46** | 56.98 | **96.88** |
| Visible+RedEdge \| NIR \| SWIR | 3 | **48.37** | **57.41** | 96.04 |
| Visible+NIR \| RedEdge \| SWIR | 3 | 47.95 | 56.74 | **97.11** |
| Visible+SWIR \| RedEdge \| NIR | 3 | 47.48 | 55.23 | 96.49 |
| Visible+SWIR \| RedEdge+NIR | 2 | 46.77 | 54.45 | 96.73 |
| Visible \| RedEdge \| NIR \| SWIR | 4 | 48.24 | **57.20** | 96.07 |

# Appendix D: Model Configurations

## D.1 MixMAE Decoder Configuration

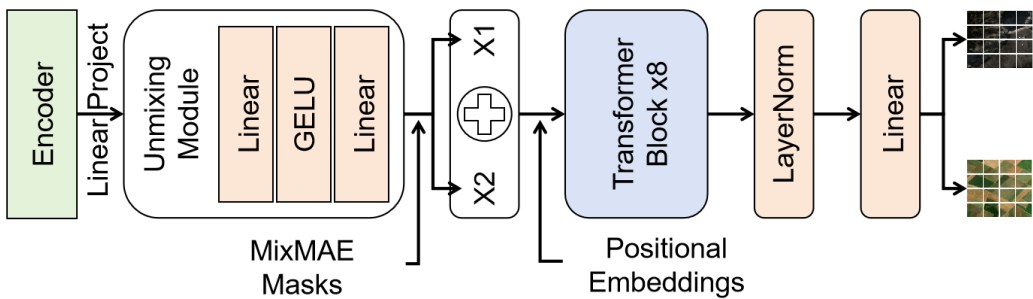

Figure 4: PhySwin Decoder Structure.

PhySwin's decoder maps the encoder's hidden states (dimension equal to the encoder's hidden size) into per-patch reconstructions via a lightweight Transformer stack, as shown in Fig. 4. Concretely, the decoder first projects hidden features into a $D_{\text{dec}} = 512$–dimensional space, then applies an *unmixing module* (two linear layers with GELU nonlinearity) to disentangle mixed tokens. Two streams are built by masking/unmasking this output according to the MixMAE mask, then concatenated and summed with bicubically-interpolated 2D sine–cosine positional embeddings. This fused sequence is processed by 8 Transformer blocks, followed by LayerNorm and a final linear prediction head that emits stride$^2 \times C$ values per token (where stride $= 4$ and $C = 10$ bands). All linear layers are Xavier-initialized and biases zeroed.

## D.2 Downstream *Plug-and-Play* Heads

To ensure fair comparisons across backbones, we adopt a "plug-and-play" evaluation strategy: all models, including PhySwins and baselines, share the same task-specific head types, and only the encoder varies across models. This design isolates the effect of pretraining and encoder quality.

Table 12: Downstream task heads and objectives. UPerNet = Unified Perceptual Parsing Network; FPN = Feature Pyramid Network; MLP = multi-layer perception. All models use a frozen encoder with plug-and-play heads.

| Dataset | Task | Head Type | Objectives |
|---------|------|-----------|------------|
| SegMunich | Semantic segmentation | UPerNet decoder (FPN + classifier) | Cross-entropy |
| Dyna.-S2 | Semantic segmentation | UPerNet decoder (FPN + classifier) | Cross-entropy |
| OSCD | Binary change detection | FPN + 3-layer Conv | Binary cross-entropy |
| Dyna.-S2 (CD) | Semantic change detection | FPN + 3-layer Conv | Cross-entropy |
| FMoW-S2 | Scene classification | GlobalAvgPool + LayerNorm + 3-layer MLP | Cross-entropy |
| EuroSAT | Scene classification | GlobalAvgPool + LayerNorm + 3-layer MLP | Cross-entropy |
| BigEarthNet | Multi-label classification | GlobalAvgPool + LayerNorm + 3-layer MLP + Sigmoid | Binary cross-entropy |

# Appendix E: Qualitative Examples

In this section, we provide additional qualitative examples for both semantic segmentation and change detection tasks. These remain highly challenging for RSFMs, and only a few of the predictions achieve acceptable accuracy. Nevertheless, PhySwin consistently delivers superior overall quality than competing methods, as seen in Figures 5, 7 and 8, and in many cases finer detail.

That said, there is still considerable room for improvement, especially on the most demanding benchmarks such as the Dyna.–S2 change detection challenge. Although PhySwin leads all SOTA

Table 13: Downstream task training configurations.

| Dataset | Input Size | Batch Size | Optimizer / LR | Epochs |
|---|---|---|---|---|
| SegMunich | 128×128 | 96 | AdamW / 5e-4 | 70 |
| Dyna.-S2 | 256×256 | 36 | AdamW / 1e-4 | 120 |
| OSCD | 96×96 | 128 | AdamW / 1e-3 | 60 |
| Dyna.-S2 (CD) | 96×96 | 64 | AdamW / 5e-4 | 70 |
| FMoW-S2 | 96×96 | 128 | AdamW / 1e-5 | 100 |
| EuroSAT | 96×96 | 128 | AdamW / 1e-3 | 100 |
| BigEarthNet | 128×128 | 96 | AdamW / 5e-5 | 100 |

baselines in the quantitative metrics reported in Table 2, its combined mask visualizations in Figure 8 remain visually inconsistent. Bridging this gap between numerical performance and perceptual fidelity will be a key focus for future RSFM development.

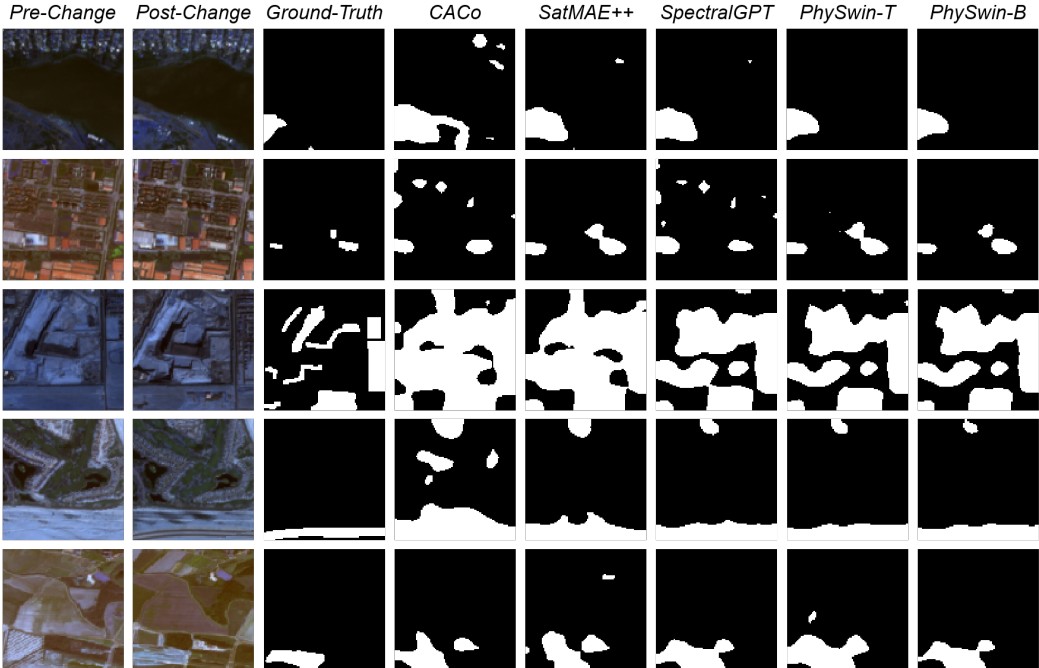

Figure 5: Additional visualizations on the OSCD dataset. Although pixel-level change detection remains challenging, PhySwin-B achieves the best performance among all baselines, producing fewer false positives.

## Appendix F: Extended Results

To facilitate reproducibility and enable fair comparison with recent Earth Observation foundation models, we integrated the PhySwin encoder into the PANGAEA evaluation framework [Marsocci et al., 2024]. We include a representative subset of benchmarks for evaluation following the official training and evaluation protocols.

**Segmentation and Change Detection.** Table 14 summarizes results on four representative benchmarks using the PANGAEA settings. Each value represents the mean ± standard deviation across three random seeds (42, 177, 892). PhySwin-B consistently ranks among the top two performers across Sentinel-2–exclusive datasets, demonstrating its strong balance between accuracy and efficiency. PhySwin-B achieves top-tier performance across multiple PANGAEA segmentation and change detection benchmarks, while PhySwin-T remains competitive given its smaller capacity. These results further support PhySwin's efficiency-oriented design.

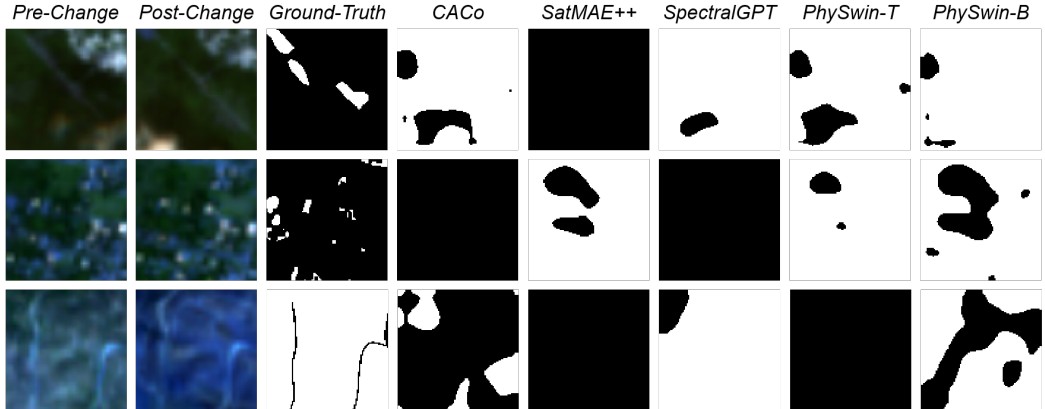

Figure 6: Additional visualizations on Dyna.-S2 change detection. This benchmark is even more challenging, and visually all RSFMs fail to produce coherent change masks. Although PhySwin-B attains the best quantitative results, these examples highlight the substantial room for improvement in RSFMs.

Table 14: PANGAEA segmentation and change detection performance (mean ± std over three seeds).

| Model | MADOS | CROPMAP | PASTIS | SEN1FLOODS11 |
|---|---|---|---|---|
| CROMA | 57.81 ± 0.34 | 44.17 ± 0.78 | 15.83 ± 0.17 | 87.12 ± 1.18 |
| DOFA | 64.40 ± 0.19 | 38.48 ± 0.53 | 12.58 ± 0.19 | 85.42 ± 0.83 |
| PRITHVI | 41.66 ± 0.70 | 50.60 ± 0.21 | 11.61 ± 0.23 | 87.38 ± 0.95 |
| DINO | 55.05 ± 0.82 | 47.93 ± 0.52 | 15.40 ± 0.18 | 86.11 ± 0.51 |
| SatLasNet | 52.80 ± 0.68 | 45.04 ± 0.36 | 13.29 ± 0.21 | 82.96 ± 1.57 |
| Terramind-L | 64.20 ± 1.07 | 49.29 ± 0.47 | 17.11 ± 0.90 | 87.45 ± 0.37 |
| PhySwin-T | 55.75 ± 0.50 | 44.37 ± 0.79 | 14.31 ± 0.14 | 85.93 ± 0.77 |
| PhySwin-B | 63.06 ± 0.88 | 51.83 ± 0.31 | 15.69 ± 0.25 | 88.30 ± 0.41 |

**Classification and F1 Metrics.** To provide a more comprehensive evaluation, we additionally report macro-F1 alongside accuracy and mean average precision (mAP) under the unified PANGAEA recipe. For EuroSAT, macro-F1 is equivalent to accuracy since the dataset is single-label and balanced. For BigEarthNet, macro-F1 is reported on the standard subset used by GeoBench (20k training and 1k testing patches). Results are shown in Table 15.

Table 15: PANGAEA classification results (mean ± std).

| Model | EuroSAT Acc | EuroSAT macro-F1 | BigEarthNet mAP | BigEarthNet macro-F1 |
|---|---|---|---|---|
| CROMA | 93.13 ± 0.18 | 93.13 ± 0.18 | 72.50 ± 0.73 | 64.90 ± 0.65 |
| DOFA | 94.85 ± 0.28 | 94.85 ± 0.28 | 69.50 ± 0.39 | 62.40 ± 0.25 |
| PRITHVI | 92.66 ± 0.85 | 92.66 ± 0.85 | 64.50 ± 0.65 | 58.30 ± 0.58 |
| DINO | 93.41 ± 1.06 | 93.41 ± 1.06 | 68.40 ± 0.37 | 60.30 ± 0.60 |
| SatLasNet | 97.81 ± 0.95 | 97.81 ± 0.95 | 70.00 ± 0.70 | 64.80 ± 0.30 |
| Terramind-L | 94.40 ± 0.19 | 94.40 ± 0.19 | 73.70 ± 0.64 | 65.73 ± 0.92 |
| PhySwin-T | 96.32 ± 0.16 | 96.32 ± 0.16 | 68.20 ± 0.68 | 61.90 ± 0.24 |
| PhySwin-B | 98.17 ± 0.19 | 98.17 ± 0.19 | 71.60 ± 0.43 | 64.10 ± 0.27 |

The unified PANGAEA results confirm that PhySwin delivers near–state-of-the-art accuracy and F1 scores while maintaining computational efficiency. Reported values are based on our own runs for PhySwin and publicly available checkpoints for other baselines.

**Ablations on Physics-Informed Losses.** To isolate the contribution of the physics-informed objectives, we conducted ablation experiments covering different pretraining variants: (i) *Physics-only* (no Spectral Group Masking or refined MixMAE), (ii) *Group-only*, (iii) *Refined-MixMAE-only*, (iv) *Full configuration (all components)*, and two baselines: a SimMIM-pretrained encoder and a

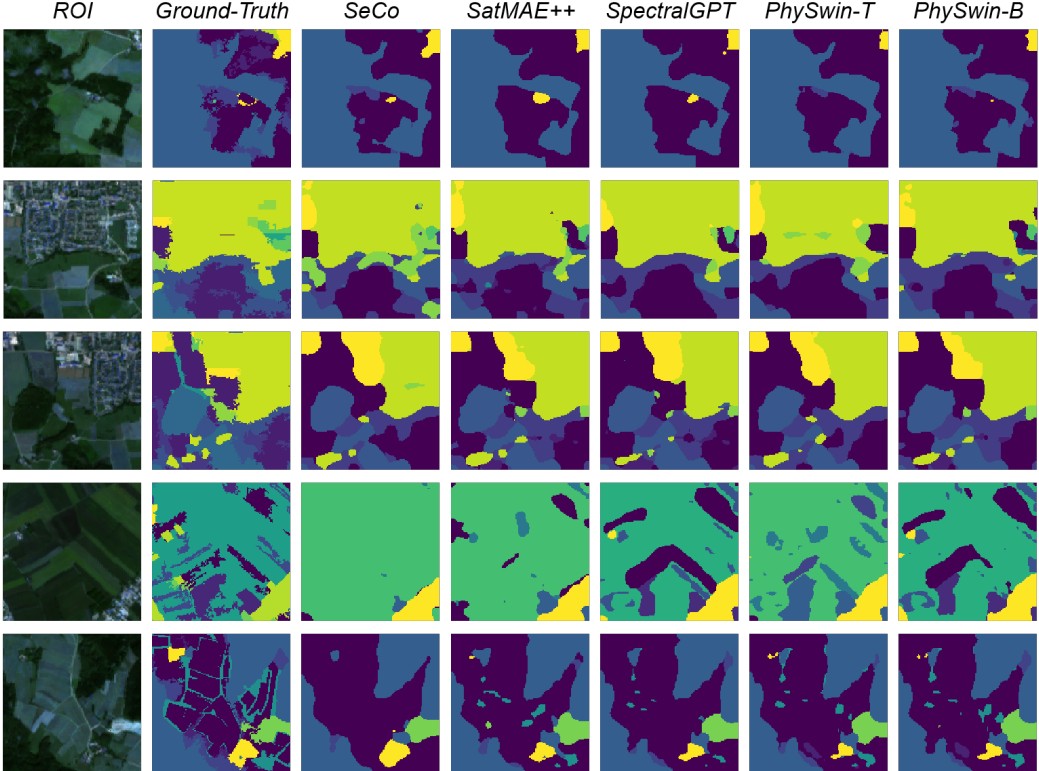

| ROI | Ground-Truth | SeCo | SatMAE++ | SpectralGPT | PhySwin-T | PhySwin-B |

Figure 7: Additional visualizations on SegMunich semantic segmentation. While most models capture the overall scene layout, PhySwin-B more accurately delineates object boundaries and small regions, demonstrating finer detail.

randomly initialized SwinV2 trained from scratch. Each variant was trained for 15 epochs using the PhySwin-T configuration and evaluated across three random seeds (42, 177, 892) to ensure statistical reliability.

Table 16: Ablation results isolating the contribution of physics-informed losses (mean $\pm$ std).

| Variant | OSCD F1 (%) | SegMunich mIoU (%) | EuroSAT F1 (%) |
|---------|-------------|--------------------|----------------|
| Base (SimMIM) | 49.31 $\pm$ 0.99 | 42.64 $\pm$ 0.85 | 92.83 $\pm$ 0.37 |
| Physics-only | 52.23 $\pm$ 0.26 | 44.71 $\pm$ 0.89 | 94.05 $\pm$ 0.94 |
| Group-only | 48.50 $\pm$ 0.49 | 40.22 $\pm$ 0.80 | 91.47 $\pm$ 1.37 |
| Refined-MixMAE-only | 49.06 $\pm$ 0.98 | 43.10 $\pm$ 0.86 | 92.61 $\pm$ 1.85 |
| Full (with all) | 52.07 $\pm$ 0.52 | 43.29 $\pm$ 0.43 | 93.73 $\pm$ 0.94 |
| Base (scratch) | 44.90 $\pm$ 0.45 | 40.37 $\pm$ 0.32 | 89.24 $\pm$ 1.34 |

Physics-only pretraining improves consistently over the SimMIM baseline across all tasks (+2.9 F1 on OSCD, +2.1 mIoU on SegMunich, and +1.2 F1 on EuroSAT), confirming that the physics-informed losses independently contribute to accuracy. The full configuration maintains these accuracy gains while providing the efficiency benefits of grouped embedding and refined MixMAE. Group-only and refined-MixMAE-only variants do not account for the observed improvements, indicating that the physics priors are the primary factor driving performance enhancements, while the architectural modifications mainly improve efficiency.

**Physics-Informed Losses on MAE Backbones.** To evaluate the generality of the proposed physics constraints beyond the Swin-based design, we further applied the same physics losses to a ViT-B backbone pretrained with standard MAE (without grouped embedding or refined MixMAE). Both

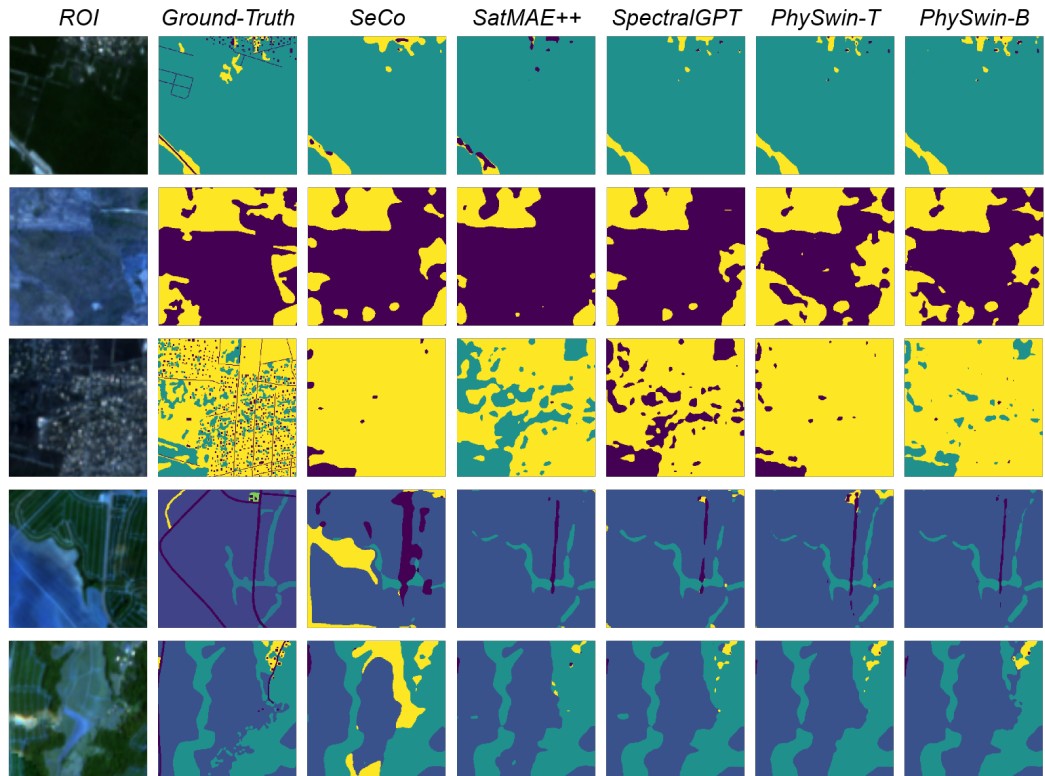

| ROI | Ground-Truth | SeCo | SatMAE++ | SpectralGPT | PhySwin-T | PhySwin-B |

Figure 8: Additional visualizations on Dyna.-S2 semantic segmentation. Despite the overall challenge, most models capture the coarse layout. PhySwin-B presents finer details.

models were trained for 15 epochs under identical settings and evaluated on OSCD, SegMunich, and EuroSAT using the same random seeds.

Table 17: Impact of physics-informed losses on MAE pretraining (mean $\pm$ std).

| Backbone | OSCD F1 (%) | SegMunich mIoU (%) | EuroSAT F1 (%) |
| --- | --- | --- | --- |
| ViT-B MAE | $50.46 \pm 0.51$ | $42.00 \pm 0.84$ | $94.78 \pm 1.90$ |
| ViT-B MAE + Physics | $52.41 \pm 0.52$ | $44.17 \pm 0.88$ | $95.20 \pm 0.95$ |

Adding physics-informed regularization consistently improves the ViT-B MAE baseline across all tasks, demonstrating the general applicability of the proposed losses. These findings align with the results reported in the main paper, showing that the spectral-smoothness and energy-conservation constraints promote inter-band consistency and suppress out-of-range reconstructions, thereby encouraging physically plausible representations and faster adaptation during downstream fine-tuning.

