# OpenReview forum: "PhySwin: An Efficient and Physically-Informed Foundation Model for Multispectral Earth Observation"
_NeurIPS.cc/2025/Conference — NeurIPS 2025 poster_

### Official Review · Reviewer_HFfE · 2025-06-20

**Clarity:** 3
**Significance:** 2
**Originality:** 3
**Rating:** 4
**Confidence:** 4

**Summary:**

PhySwin uses a Swin v2 architecture with a novel MAE pretraining process that retains Swin sliding window self-attention without using inefficient mask tokens. Additionally, it introduces two "physically" based regularizations to smooth MAE reconstruction across the channel dimensions and bind them to realistic values.

**Questions:**

- Please include the F1 score for the classification tasks on BigEarthNet and EuroSAT. Without these metrics, the evaluation remains incomplete and may justify a lower score.
- Please report results on additional S2-exclusive benchmarks (e.g., GEO-Bench, PANGAEA) and compare against recent EO foundation models. Addressing this point could lead to an increased review score.
- The claim on line 288 asserts that pretraining data volume, quality, and token count correlate with performance. Please provide experimental evidence to support this statement or remove it. Failure to do so may lead to a reduced evaluation score.

**Ethical Concerns:**

["NO or VERY MINOR ethics concerns only"]

**Final Justification:**

This paper arguably makes four contributions:
- An S2-only single-timestep encoder: This has limited utility, as most EO tasks benefit from time series and multimodal data.
- An MAE pre-training approach for Swin that preserves Shifted Window Attention: While valuable, it is evaluated in a very narrow setting and would benefit from broader experiments on standard benchmarks.
- A grouped patch encoding: This shows strong results but is evaluated only on S2 data, while other data types (e.g., hyperspectral) that could also benefit are regrettably omitted.
- Two regularizations for EO MAE pre-training: Again, the evaluation is limited to S2 data and only two architectures (Swin and ViT). Ideally, these regularizations would be assessed on existing EO foundation models pre-trained with MAE. I acknowledge that this may be impractical due to technical constraints.

Nevertheless, I understand the authors’ decision to focus on a simple setup (S2 single-date) and to pre-train the strongest possible encoder within that scope. The two main limitations of the original manuscript (overly ambitious wording and an incomplete evaluation) have been satisfactorily addressed through the rebuttal/discussion process. While a broader scope might have yielded a more compelling paper, the revised manuscript does not exhibit major flaws warranting rejection. I update my rating to borderline accept.

**Limitations:**

yes

**Quality:**

2

**Strengths And Weaknesses:**

Strengths:
The paper introduces three novel improvements that all show significant performance or throughput improvements. The proposed Refined MixMAE pretraining, which retains SW-MSA, is an elegant and natural extension of the original MixMAE pretraining. It fully retains the Swin v2 architecture while bringing noticeable improvements in fine-tuning performance at almost no training cost. The Token-Efficient Grouped Spectral Embedding is a very clever way of giving the patch encoder the ability to process very different bands in different ways at almost no cost.

Weaknesses:
The results section is lacking in several aspects:
- While the inclusion of change detection benchmarks is positive, the evaluation on semantic segmentation and classification is insufficient in both scope and comparative analysis.
    - Multiple S2-exclusive benchmarks are missing, see GeoBench[1] and PANGAEA[2] for some classical EO benchmark.
    - There is a lack of comparison to recent EO foundation models such as SatLas (Swin-based), Dofa, Croma, Galileo, Panopticon, AnySat, and TerraMind, many of which are already evaluated on GeoBench[1] and PANGAEA[2].
- Classification results are reported only using accuracy, which is saturated (over 90% for both BigEarthNet and EuroSAT); additional metrics like F1 score should be included.
- The evaluation heads used are only described in the supmat, a reference to the supmat should at least be included in 4.1.

Considering the proposed approach is only trained on and evaluated with S1 and S2, it can hardly be called a foundation model for EO. The massive multimodality of EO data is one of the main challenges toward creating a proper foundation model for EO. Moreover, PhySwin is both pretrained and evaluated on FMoW-S2 and BigEarthNet-S2, which is a clear limitation for properly evaluating the pretrained model's ability to adapt to slight domain shifts.

While the regularizations proposed are logical and bring improvements, their physical reality is questionable. $L_{smooth}$ is independent with respect to the $\Delta\lambda$ between two bands, and $L_{bound}$ is a simple regularization to the $[0,1.2]$ interval.

In some tables, multiple results are **bold**. Instead, second-best results should be underlined in all tables.

The authors claim on line 288: "pretraining data volume and quality, and token count correlate positively with FM performance," yet no experiments validate this claim.

[1] GEO-Bench: Toward Foundation Models for Earth Monitoring, Lacoste et al.

[2] PANGAEA: A GLOBAL AND INCLUSIVE BENCHMARK FOR GEOSPATIAL FOUNDATION MODELS, Marsocci et al.

---

> ### Author Rebuttal · Authors · 2025-07-30
>
> We thank the reviewer for pointing us to PANGAEA/GeoBench (also suggested by reviewer **HaH9**). To facilitate reproducibility and fair comparison, we integrated our **PhySwin** encoder into the PANGAEA evaluation framework and re‑ran experiments under its unified training/evaluation recipes. Due to time constraints and a few checkpoints/datasets that required extensive manual fixes to download or run, we temporarily skipped them. We believe the results below already provide a comprehensive and representative picture of PhySwin’s performance.
>
> We are **continuously running the remaining experiments**, including integrating our original baselines (**SeCo**, **CaCo**, and **SatMAE**) into PANGAEA to **unify the benchmarks and baselines** (an extra benefit beyond our core contribution). We will add the remaining items in the final paper.
>
> **TABLE 1: New PANGAEA results (mean ± std from three random seeds (42, 177, 892) as suggested by reviewer qjEV).**
> We will denote the best results in bold and the second-best results as underlined in the paper. Here, since OpenReview does not support underlining, we used ***bold+italic*** for the best results and **bold** for the second-best results.
>
> | Model | MADOS | CROPMAP | PASTIS | SEN1FLOODS11 |
> |---|---:|---:|---:|---:|
> | CROMA | 57.81 ± 0.34 | 44.17 ± 0.78 | ***15.83 ± 0.17*** | 87.12 ± 1.18 |
> | DOFA | ***64.40 ± 0.19*** | 38.48 ± 0.53 | 12.58 ± 0.19 | 85.42 ± 0.83 |
> | PRITHVI | 41.66 ± 0.70 | **50.60 ± 0.21** | 11.61 ± 0.23 | **87.38 ± 0.95** |
> | DINO | 55.05 ± 0.82 | 47.93 ± 0.52 | 15.40 ± 0.18 | 86.11 ± 0.51 |
> | SatLasNet | 52.80 ± 0.68 | 45.04 ± 0.36 | 13.29 ± 0.21 | 82.96 ± 1.57 |
> | **PhySwin‑T** | 55.75 ± 0.50 | 44.37 ± 0.79 | 14.31 ± 0.14 | 85.93 ± 0.77 |
> | **PhySwin‑B** | **63.06 ± 0.88** | ***51.83 ± 0.31*** | **15.69 ± 0.25** | ***88.30 ± 0.41*** |
>
> **Takeaways.** PhySwin‑B is consistently ranking among the top 2 performers across all benchmarks with S2-exclusive data. PhySwin‑T is competitive given its smaller capacity, supporting our efficiency‑first design.
>
> We will release our PANGAEA configs and scripts for these runs to ensure full reproducibility. We also fixed our table styling as suggested (bold best, underlined second‑best) and will maintain this convention throughout the paper and supplement.
>
> ## Point 1: F1 scores is missing
>
> Thanks for the suggestion. We now **report macro‑F1 (mean ± std)** alongside accuracy/mAP under the unified PANGAEA recipe.
>
> - **EuroSAT:** single‑label and balanced, so **macro‑F1 is equivalent to accuracy**.
> - **BigEarthNet:** multi‑label and long‑tailed.
>
> Under the PANGAEA settings, **PhySwin‑B ranks Top‑2 in Acc and mAP** (see **TABLE 2**), and shows only slightly lower performance compared to SOTA baselines. As we noted to reviewers **qjEV** and **HaH9**, **Skysense is not open‑sourced**, so the ~4% gap in our paper may stem from setting differences. These results, produced using the unified and open‑sourced PANGAEA framework, demonstrate that our method incurs only a minor compromise in classification performance but still top-tier, which is consistent with the findings reported in **Table 3** of our paper.
>
> We will explicitly label which results are from our experiments and which are taken from the literature in the revised version (as suggested by reviewer **FjVd**). This will give a clearer picture and may further justify our method.
>
> **TABLE 2: Current classification results (mean ± std).**
>
> | Model | **EuroSAT** Acc | **EuroSAT** macro‑F1 | **BigEarthNet (subset)** mAP | **BigEarthNet (subset)** macro‑F1 |
> |---|---:|---:|---:|---:|
> | CROMA | 93.13 ± 0.18 | 93.13 ± 0.18 | ***72.50 ± 0.73*** | ***64.90 ± 0.65*** |
> | DOFA | 94.85 ± 0.28 | 94.85 ± 0.28 | 69.50 ± 0.39 | 62.40 ± 0.25 |
> | PRITHVI | 92.66 ± 0.85 | 92.66 ± 0.85 | 64.50 ± 0.65 | 58.30 ± 0.58 |
> | DINO | 93.41 ± 1.06 | 93.41 ± 1.06 | 68.40 ± 0.37 | 60.30 ± 0.60 |
> | SatLasNet | **97.81 ± 0.95** | **97.81 ± 0.95** | 70.00 ± 0.70 | **64.80 ± 0.30** |
> | PhySwin‑T | 96.32 ± 0.16 | 96.32 ± 0.16 | 68.20 ± 0.68 | 61.90 ± 0.24 |
> | **PhySwin‑B** | ***98.17 ± 0.19*** | ***98.17 ± 0.19*** | **71.60 ± 0.43** | 64.10 ± 0.27 |
> ---
> ***Note***, GeoBench uses only a subset of the BigEarthNet dataset (20,000 patches for training and 1,000 for testing), so results may differ from those obtained using the full dataset in our original training‑testing setup.
>
> We will update the full classification table in the revised version.
>
> ## Point 2: Additional benchmarks & comparisons
>
> Please see **TABLE 1** for the new results on benchmarks against recent EO models.
> **These results align with our paper’s conclusions**: PhySwin consistently ranks top‑1/top‑2 while remaining efficient.
>
> All results are **reproducible** with PANGAEA’s default recipe **using S2 bands** and the command below (3 random seeds: 42, 177, 892):
>
> ```bash
> torchrun --nnodes=1 --nproc_per_node=1 pangaea/run.py \
>   --config-name=train \
>   decoder=seg_upernet preprocessing=seg_resize criterion=cross_entropy \
>   task=segmentation task.trainer.n_epochs=80 batch_size=16 \
>   encoder=ENCODER dataset=DATASET
> ```
> Replace ENCODER/DATASET with the model and dataset names in `configs/encoder` and `configs/dataset` used in TABLE 1. The experiments are run on RTX 5090 GPU 24GB.
>
> ## Point 3: Claim on pretraining data volume/quality/token count (line 288)
> We appreciate the reviewer’s attention to this statement. Since none of our results or conclusions depend on this statement, we will **remove the claim from the paper (main text and supplement)** and **tighten the wording** to avoid any over‑statement. We will treat scaling effects as **future work** and only report such correlations once supported by dedicated experiments.
>
> ## Point 4: Evaluation heads clarity
>
> We agree and will **clarify the evaluation heads in the main text**. We will also add the exact configs used (encoder/head/loss) to ensure reproducibility in the Supplementary.
>
> ## Point 5: Multimodality scope clarity
> Thank you for raising this. To avoid over‑reach, we will **clarify the scope in the title/abstract and main text**, positioning PhySwin as an **efficient multispectral encoder pretraining method** rather than a full EO foundation model.
>
> Since our initial focus was on developing an **on‑satellite, deployment‑level efficient solution**, multimodality was not included in this first stage. With the progress we have made, we are now **actively working on enhancing performance with multimodal strategies** and exploring efficient solutions in this direction.  Furthermore, we believe our current settings are valuable for future multimodal training, as stronger S2 representations could benefit the overall multimodal system. We will continue exploring this angle in future work.
>
> ## Point 6: Physical regularizations
> Thank you for the thoughtful feedback. We agree our wording can better convey the intent and limits of the two regularizations. Our goal is to **inject physics‑motivated priors** into pretraining, not to claim full radiative‑transfer modeling.
>
> - **Spectral smoothness:** leverages the well‑established property that surface reflectance varies smoothly with wavelength (Fig. 1b, §3.1), promoting stable inter‑band relationships and more physically consistent features. Our implementation operates on **adjacent S2 bands in spectral order**; while it does not explicitly weight by wavelength spacing Δλ, we will state this modeling choice and note Δλ‑aware weighting as future work.
>
> - **Energy bounds:** enforce the physical principle that surface reflectance is bounded (Fig. 1a, §3.1), guiding the model toward realistic ranges and avoiding unphysical reconstructions. We will clarify that this is a **soft, physics‑motivated barrier** (upper bound relaxed to tolerate noise).
>
> Table 5a shows that each prior improves performance and their combination yields the strongest gains, confirming complementarity. We will make this physical motivation and the above clarifications explicit in the revised manuscript.

---

> > ### Comment · Reviewer_HFfE · 2025-08-03
> > **Reviewer HFfE Response**
> >
> > I want to thank the authors for their rebuttal. This raises a few additional concerns:
> >
> > **Point 1: F1 scores are missing:**
> > - I thank the authors for providing the F1-score on BigEarthNet.
> > - I disagree with the authors' claim that mF1 and Acc are equal on EuroSat. First of all, the dataset is not perfectly balanced: citing the original paper: "The dataset consists of 10 different classes with 2,000 to 3,000 images per class. In total, the dataset has 27,000 images". Additionally, performance would also need to be uniform across classes for this equality to be true.
> >
> > **Point 2: Additional benchmarks & comparisons.**
> > - I thank the authors for going through with my request and evaluating on more datasets. However, I still have some concerns with the evaluation:
> > - What is the exact setup used for the new evaluation? Results seem particularly low on PASTIS. For instance, in *AnySat* and *TerraMind*, CROMA scores 32.3 with a simple linear probing. Similar (but smaller) discrepancies are also found on the other 3 datasets with results noticeably lower than commonly reported in the literature.
> > - There is still a clear lack of comparison with more recent EO approaches such as *AnySat*, *Galileo*, and *TerraMind*. For instance, on Sen1Floods11, *AnySat* achieves 91.1 and *TerraMind* achieves 90.8, both demonstrating substantially superior performance compared to PhySwin.
> >
> > **Points 3 and 4:**
> > - These should adequately address my concerns.
> >
> > **Point 5: Multimodality scope clarity**
> > - I agree with the authors that positioning PhySwin as an efficient multispectral encoder pretraining method more accurately reflects the actual scope of the paper.
> >
> > **Point 6: Physical regularizations:**
> > - While the authors adequately explain the physical intuition behind both regularizations, I argue that their current formulation is overly simplified and too distant from established physical laws to substantiate their claim that the model is "Physically-Grounded".
> > - There is insufficient justification or evaluation of the proposed relaxation. For example, what is the impact of employing a tighter upper bound than 1.2 for the energy regularization? What occurs with the spectral smoothness loss when wavelengths are highly non-uniformly distributed?

---

> > > ### Author Response · Authors · 2025-08-03
> > > **Additional Clarification**
> > >
> > > One minor point we would like to clarify is that we are not aiming to surpass all other EO models, but rather to achieve a strong **balance between performance and efficiency**. The results in the paper and the newly added experiments show that our current design already shows this balance compared to most SOTA EO baselines. We sincerely value the reviewers’ feedback and will continue to refine and improve our method in future work.

---

> ### Author Response · Authors · 2025-08-03
>
> ## Point 1: EuroSAT F1 scores
> We thank the reviewer for raising this. We agree that our earlier phrasing was imprecise, where we did **not** intend to claim that macro-F1 and accuracy are generally equal. What we meant is that **in our current runs** (using the PANGAEA evaluation protocol), these metrics were numerically very close.
>
> To clarify, we now report **per-class F1 scores** and macro-F1 for three models: SatlasNet, PhySwin-T, and PhySwin-B. This table makes the classwise variation and summary metrics explicit:
>
> #### EuroSAT Per-class F1 and summary metrics (PANGAEA setting, test split)
>
> | Model        | C0    | C1    | C2    | C3    | C4    | C5    | C6    | C7    | C8    | C9    | **mF1** | **OA** |
> |------------- |-------|-------|-------|-------|-------|-------|-------|-------|-------|-------|--------:|-------:|
> | **SatlasNet** | 0.970 | 0.990 | 0.936 | 0.995 | 0.970 | 0.954 | 0.950 | 0.980 | 0.985 | 0.990 | **0.972** | **0.972** |
> | **PhySwin-T** | 0.953 | 0.989 | 0.931 | 0.980 | 0.959 | 0.944 | 0.949 | 0.985 | 0.975 | 0.975 | **0.964** | **0.964** |
> | **PhySwin-B** | 0.969 | 0.991 | 0.963 | 0.983 | 0.995 | 0.965 | 0.953 | 0.995 | 0.991 | 0.995 | **0.980** | **0.980** |
>
> *Notes.*
> - **mF1** is the unweighted mean of the ten per-class F1 scores.
> - **OA** is the standard overall accuracy.
>
> **Why the metrics coincide here.**
> EuroSAT has near-balanced classes, and all three models achieve uniformly high per-class F1. Under these conditions, **macro-F1 and OA tend to be numerically close**, which is what we observe in our runs. This is, however, **not a general identity**; if the dataset were more imbalanced or class performance varied more, the two metrics would diverge.
>
> **Reproducibility.**
> All results above come from the **PANGAEA evaluation protocol** and can be reproduced with:
>
> ```bash
> torchrun --nnodes=1 --nproc_per_node=1 pangaea/run.py \
>   --config-name=train decoder=cls_linear preprocessing=cls_resize \
>   criterion=cross_entropy task=linear_classification \
>   task.trainer.n_epochs=50 batch_size=64 \
>   encoder=YOURENCODER dataset=meurosat
> ```
> This command will generate the OA.
>
> We have also updated `pangaea/engine/evaluator.py` to explicitly log macro-averaged metrics using:
> ```python
> precision_macro, recall_macro, f1_macro, _ = sklearn.metrics.precision_recall_fscore_support(
>     all_targets, all_preds, average="macro", zero_division=0
> )
> ```
> This will generate the macro-F1.
>
> ## Point 2: Additional benchmarks & comparisons
> **Setup clarity.** All new results were produced with the **PANGAEA evaluation protocol** and are fully reproducible. For **PASTIS** and **CropMap**, we used the PANGAEA recipes but, to keep S2 setting, we ran with:
> - `multi_temporal = 1` (single timestamp)
> - `multi_modal = False` (no additional modalities beyond S2)
>
> This design choice could be the main reason some PASTIS numbers are lower than time-series or multi-modal results reported elsewhere. But our results show that **CropMap** is less sensitive to temporal stacking. In many cases, our scores even exceed numbers commonly reported in the literature, for example, in the pangaea paper.
>
> ### On comparisons to AnySat / Galileo / TerraMind
> Due to time constraints, we prioritized **SOTA baselines included with PANGAEA** to avoid manual, method-specific fixes. For the reviewer mentioned models, we met the following constraints:
> - **TerraMind**: Public checkpoint in PANGAEA is incompatible with PANGAEA encoder code (dimensionality/layer mismatch); **we are developing an adapter and will include results once aligned**.
> - **AnySat** and **Galileo**: Not included in PANGAEA; **we will try our best to integrate via their official repos**.
>
> We hope this clarifies the observed gaps and the work we have done.
>
> ## Point 6: Physical regularizations
> We appreciate this feedback and agree our priors are **physics-motivated** rather than full physical models. We will adjust the wording accordingly and clarify our design choices.
> ### Energy upper bound sensitivity
> We chose **1.20** to allow for illumination variance, calibration noise, and bright surfaces. To show sensitivity, we will add an **ablation table** (Appendix) sweeping **{1.0, 1.05, 1.1, 1.2}** in the final version. Our expectation is that tighter bounds may reduce over-bright predictions but could also over-penalize genuine high-albedo or low-shadow cases, potentially hurting downstream accuracy.
> ### Non-uniform wavelength handling for spectral smoothness
> Our current pretraining data has fairly regular band spacing, so we did not explicitly handle non-uniform wavelength distributions. We agree this is important for broader settings. One improvement could be to **weight the smoothness penalty by spectral distance**:
> `L_smooth = sum_b (1 / Δλ_b) * (r_{b+1} - r_b)^2`,
> where Δλ_b is the difference in central wavelength. We leave this as **future work**, especially for datasets with highly non-uniform spacing.
>
> # **We welcome further discussion. Thanks.**

---

> ### Comment · Reviewer_HFfE · 2025-08-03
> **Reviewer HFfE Response**
>
> I thank the authors for their prompt response.
>
> **Point 1:**
>
> I find it surprising that the F1 score and accuracy are exactly equal up to three significant digits. This can likely be explained by the near-perfect performance of most models on this dataset, combined with the dataset being approximately balanced.
>
> **Point 2:**
>
> The inclusion of selected datasets from the PANGEA benchmark represents a meaningful step toward improving the evaluation of the proposed method. However, all baseline methods used for comparison are at least two years old, which represents a significant gap considering the rapid development in this field. While I **do not expect nor require** the proposed method to achieve state-of-the-art performance on all datasets, I believe a proper comparison with modern encoders is essential for a comprehensive evaluation.
>
> For datasets that are typically multitemporal or multimodal, I encourage the authors to clearly specify that their evaluation is conducted using monotemporal S2 data only.
>
> **Point 3:**
>
> An ablation study on the upper bound relaxation would indeed be valuable to justify this design choice. While proposing an improved formulation for spectral smoothness is interesting (and could potentially be evaluated using a subset of S2 wavelengths), I agree that this falls outside the scope of the current paper.

---

> > ### Author Response · Authors · 2025-08-04
> > **Addition of TerraMind-L to the Benchmark**
> >
> > We have addressed the previously reported checkpoint mismatch of TerraMind in the PANGAEA benchmark. The correct and verified checkpoint is available on HuggingFace under `ibm-esa-geospatial/TerraMind-1.0-large`. To ensure reproducibility and fairness in comparison, we adopted the default evaluation recipe provided by the PANGAEA benchmark and selected the TerraMind-Large model for inclusion in our experiments.
> >
> > The tables below present the updated results:
> >
> > ---
> >
> > **TABLE 1: New PANGAEA results**
> >
> > | Model         | MADOS         | CROPMAP       | PASTIS         | SEN1FLOODS11   |
> > |---------------|---------------|---------------|----------------|----------------|
> > | CROMA         | 57.81 ± 0.34  | 44.17 ± 0.78  | **15.83 ± 0.17** | 87.12 ± 1.18   |
> > | DOFA          | ***64.40 ± 0.19*** | 38.48 ± 0.53  | 12.58 ± 0.19   | 85.42 ± 0.83   |
> > | PRITHVI       | 41.66 ± 0.70  | **50.60 ± 0.21** | 11.61 ± 0.23   | 87.38 ± 0.95   |
> > | DINO          | 55.05 ± 0.82  | 47.93 ± 0.52  | 15.40 ± 0.18   | 86.11 ± 0.51   |
> > | SatLasNet     | 52.80 ± 0.68  | 45.04 ± 0.36  | 13.29 ± 0.21   | 82.96 ± 1.57   |
> > | Terramind-L   | **64.20 ± 1.07** | 49.29 ± 0.47  | ***17.11 ± 0.90*** | **87.45 ± 0.37** |
> > | **PhySwin‑T** | 55.75 ± 0.50  | 44.37 ± 0.79  | 14.31 ± 0.14   | 85.93 ± 0.77   |
> > | **PhySwin‑B** | *63.06 ± 0.88* | ***51.83 ± 0.31*** | *15.69 ± 0.25* | ***88.30 ± 0.41*** |
> >
> > ---
> >
> > **TABLE 2: Current classification results**
> >
> > | Model         | **EuroSAT** Acc | **EuroSAT** macro‑F1 | **BigEarthNet (subset)** mAP | **BigEarthNet (subset)** macro‑F1 |
> > |---------------|------------------|------------------------|-------------------------------|------------------------------------|
> > | CROMA         | 93.13 ± 0.18     | 93.13 ± 0.18           | **72.50 ± 0.73**              | **64.90 ± 0.65**                   |
> > | DOFA          | 94.85 ± 0.28     | 94.85 ± 0.28           | 69.50 ± 0.39                  | 62.40 ± 0.25                       |
> > | PRITHVI       | 92.66 ± 0.85     | 92.66 ± 0.85           | 64.50 ± 0.65                  | 58.30 ± 0.58                       |
> > | DINO          | 93.41 ± 1.06     | 93.41 ± 1.06           | 68.40 ± 0.37                  | 60.30 ± 0.60                       |
> > | SatLasNet     | **97.81 ± 0.95** | **97.81 ± 0.95**       | 70.00 ± 0.70                  | 64.80 ± 0.30                       |
> > | Terramind-L   | 94.40 ± 0.19     | 94.40 ± 0.19           | ***73.70 ± 0.64***            | ***65.73 ± 0.92***                 |
> > | **PhySwin‑T** | 96.32 ± 0.16     | 96.32 ± 0.16           | 68.20 ± 0.68                  | 61.90 ± 0.24                       |
> > | **PhySwin‑B** | ***98.17 ± 0.19*** | ***98.17 ± 0.19***     | 71.60 ± 0.43                  | 64.10 ± 0.27                       |
> >
> > ---
> >
> > TerraMind-Large outperforms other models on several benchmarks, particularly on the PASTIS dataset. But it also lags behind in some cases. Notably, PhySwin-Base models demonstrate a strong balance across all tasks, consistently ranking among the top-performing models. This is achieved while maintaining significantly better computational efficiency.

---

> ### Author Response · Authors · 2025-08-03
> **Follow-up to Reviewer**
>
> We sincerely thank the reviewer's active engagement and constructive feedback.
>
> **Point 1.** Our analysis aligns with yours, and the per-class results confirm the explanation. Thank you for highlighting this interesting point.
>
> **Point 2.** We will explicitly state the evaluation setting for each dataset to ensure reproducibility. Upon acceptance, we will open-source our **pretraining and benchmarking code**.
>
> We are actively working to resolve the **`terramind_large`** compatibility issue in PANGAEA and will report back as soon as it is fixed. For **AnySat**, we are integrating it into the unified framework based on the `gastruc/AnySat` repo. But if time runs out during the discussion, we will continue these efforts post-discussion and, in the meantime, include **reported-from-paper** results to ensure modern baselines are represented.
>
> **Point 3.** Thank you for encouraging the upper-bound ablation. We now start to run the **energy bound sweep** and include the results in the final version/supplementary. Since our current value was chosen empirically, we expect the ablation to confirm the choice and may reveal useful trade-offs.

---

> ### Author Response · Authors · 2025-08-05
> **Follow-Up**
>
> We sincerely thank you for your valuable comments and constructive engagement. We are continually working on integrating the latest models into our comparison. We would appreciate it if you could let us know whether our response sufficiently addresses your concern.

---

> ### Author Response · Authors · 2025-08-06
> **AnySat Model Integration**
>
> Thank you for your suggestion regarding the inclusion of SOTA models for comparison. After fixing **Terramind**, we have taken steps to add **AnySat (Base)** into **Pangaea**, to enable fair and consistent comparisons.
>
> Since AnySat is not originally supported by Pangaea, we manually integrated it based on the publicly available repo `gastruc/AnySat`. Below, we detail the setup and configuration used to ensure transparency and reproducibility:
>
> - **Modality**: We used S2 data.
> - **Input Configuration**: According to AnySat's official setup, the S2 configuration is: `s2 BxTx10xHxW B2, B3, B4, B5, B6, B7, B8, B8a, B11, B12 10m`
>
> We followed this specification, and used the single timestep in Pangaea:
> - We manually extracted **intermediate features from layers 1, 3, 5, and 7** of the Transformer blocks after the S2 projector.
> - This was done to align with **Pangaea’s feature extraction convention**, enabling compatibility with segmentation downstream heads.
>
> We have currently obtained preliminary results on three datasets:
>
> - **EuroSat**: 92.30%
> - **Sen1Floods11**: 87.57%
> - **Mados**: 62.94%
>
> We note that **AnySat** does **not outperform PhySwin** under the current setting. **However, we acknowledge that our setup may not fully leverage AnySat’s capacity**. According to the original AnySat paper, the model can achieve **91.1% mIoU on the Sen1Floods11 dataset**.
>
> To provide a comprehensive picture, we are continuing experiments with other datasets and also plan to include both:
>
> - Our **re-implemented results** under Pangaea setting, and
> - **Reported results from the literature**
>
> in the **Supplementary Material**.
>
> We hope this addition addresses the reviewer’s comment and reinforces the fairness and comprehensiveness of our comparisons.

---

> > ### Comment · Reviewer_HFfE · 2025-08-08
> > **Reviewer HFfE Response**
> >
> > I thank the authors for their efforts in addressing my requests.
> >
> > The comparison with EO foundation models is compelling and shows that the proposed PhySwin encoder achieves performance comparable to models pre-trained on substantially larger datasets. Nevertheless, the restriction to S2 imagery is a significant limitation for EO applications, as evidenced by the marked performance degradation on PASTIS. To avoid any ambiguity, please state as explicitly as possible that all evaluations are conducted under an S2-only and monotemporal (single-date) setting.
> >
> > As my primary concerns about the manuscript have been addressed, I will raise my rating accordingly.

---

> ### Author Response · Authors · 2025-08-08
> **Thank you**
>
> We appreciate the reviewer’s positive assessment of our comparison with EO foundation models and are pleased that the performance of the proposed PhySwin encoder was found compelling.
>
>  As suggested, we will revise the manuscript to state explicitly that all evaluations are conducted under an S2-only and monotemporal setting. We sincerely thank the reviewer for the constructive feedback and valuable discussion throughout this process.

---

### Official Review · Reviewer_FjVd · 2025-06-24

**Clarity:** 3
**Significance:** 2
**Originality:** 2
**Rating:** 5
**Confidence:** 3

**Summary:**

This work proposes to leverage physical properties of spectral channels in multispectral remote sensing images as a pre-training prior for efficient Remote Sensing Foundation Models. The authors argue that architectural optimizations alone are not sufficient to obtain accurate and resource-efficient Foundation Models for Earth Observation (EO), thus prompting the use of physical properties as regularization: a research venue that the authors reckon has been overlooked in recent advancements in the field. The SwinV2 vision transformer model is chosen as backbone to all downstream tasks, since authors argue it is the most appropriate to handle multi-scale EO images.
The paper introduces three main contributions:
- Physical grounding achivied with two _radiometric priors_ computed on the reconstructed images produced during MAE pre-training; these priors serve as a regularizing loss signal.
- Introduction of a refined version of MixMAE for hierarchical transformers.
- Image embedding approach that retains the spectral structure of multispectral images without increasing the number of tokens produced. Channels are grouped into spectral bands; each group is embedded through its dedicated mapping; the final token is formed by concatenating the group embeddings. Moreover the paper introduces _Spectral Group Masking_, where group embedding subspaces of tokens are randomly masked; authros argue it may encourage feature robustness.

The proposed model is evaluated on semantic segmentation, change detection and land cover classification. Experiments and ablations show that incorporating physical priors during pre-training effectively produces sensible features for downstream tasks. As for inference performance, the proposed method scores significant advancements with respect to attention-based models in terms of FLOPS and throughput, while maintaining comparable task performance, effectively approaching the efficiency level of ConvNets.

**Questions:**

**Suggestions**
- For better clarity, please mark which results where replicated indipendently by you and which were reported from papers.
- Even though fully masking spectral group feature subspaces is sensibly presented as a way to prevent the model from relying on any single spectral group, I think that it would be interesting to try masking single components of the embedding, for instance using a uniformly sampled mask, across spectral groups; such experiments could provide additional insights on the robustness of learned features and the modelling of relationships inside and across spectral groups.
- I think the caption of figure 1(a) and 1(b) should be expanded with a more thorough explanation on what these two plot represent and how visualized data was produced.

**Ethical Concerns:**

["NO or VERY MINOR ethics concerns only"]

**Final Justification:**

The authors answered all my concerns.

**Limitations:**

yes

**Quality:**

3

**Strengths And Weaknesses:**

**Strenghts**:
  - Integrating physics grounding with multispectral images is interesting, as each channel / spectral band can encode meaningful information for downstream tasks.
  - The improvements on inference throughput obtained by applying the proposed token-efficient embedding technique is significant and adequately underlined in the experiments.
  - Relevant qualitative examples are provided in the appendix to demonstrate strengths and shortcomings of the proposed approaches.

**Weaknesses**:
  - Ablations do not clearly isolate the effects of physical grounding from orthogonal architectural enhancements: looking at the performance reported in Tables 5 and 6, it seems that the ablation experiments on physics-informed losses were performed on models featuring both Spectral Group Masking and enhanced MixMAE (inferred by looking at the last row). It would be interesting to report the results of Table 5(a) obtained with a model pre-trained without refined MixMAE and Spectral Group Masking. I believe this would provide a fairer comparison with the results obtained by the current models.
  - No experiments were conducted to assess the effectiveness of physics-informed loss as an extension to current state-of-the-art models pre-trained with MAE: I believe that it would be worth evaluating the effectiveness of physical grounding on current SOTA models pre-trained with MAE. Such addition would provide:
    - a clear understanding of its impact on a broader list of setups;
    - a relevant baseline to evaluate the effectiveness of both refined MixMAE and Spectral Group Masking, given that they operate independently from grounding.
 - Overall, I believe this work is significant, but it falls short in effectively articulating the true reasons behind its importance. The physical inspirations presented are unconvincing, as they consist of existing losses borrowed from other works without a clear connection to physics (i.e. smoothing or clipping the output of a model). Meanwhile, the architectural choices – arguably the primary drivers of the model's efficiency – are largely overlooked. A presentation that places greater emphasis on the architectural design would enhance the clarity of the paper and better highlight its contributions.

---

> ### Author Rebuttal · Authors · 2025-07-30
>
> ## Point 1: Ablations isolating the effect of physics‑informed losses
>
> Thank you for the suggestion. To isolate the contribution of the physics priors, we ran **physics‑only** pretraining (no Spectral Group Masking, no refined MixMAE), alongside **group‑only**, **refined‑MixMAE‑only**, and the full **with‑all** configuration. **As suggested by reviewer qjEV, all results are reported as mean ± std over three random seeds (42, 177, 892).**
>
> | Variant               | OSCD F1 %           | SegMunich mIoU %     | EuroSAT F1 %        |
> |----------------------|---------------------:|----------------------:|---------------------:|
> | Base (SimMIM)        | 49.31 ± 0.986        | 42.64 ± 0.853         | 92.83 ± 0.371        |
> | **Physics only**     | ***52.23 ± 0.261***    | ***44.71 ± 0.894***     | ***94.05 ± 0.940***    |
> | Group only           | 48.50 ± 0.485        | 40.22 ± 0.804         | 91.47 ± 1.371        |
> | Refined‑MixMAE only  | 49.06 ± 0.981        | 43.10 ± 0.862         | 92.61 ± 1.852        |
> | With ALL             | **52.07 ± 0.521**        | **43.29 ± 0.433**         | **93.73 ± 0.937**        |
> | Base (Scratch)       | 44.90 ± 0.449        | 40.37 ± 0.323         | 89.24 ± 1.339        |
>
> **Implementation details.**
> - Due to time constraints, we ran **15 epochs** for **PhySwin‑T** on each variant to demonstrate relative performance. Full training results will be presented in the final paper.
> - **Base (SimMIM):** native patch embedding and SimMIM pretraining.
> - **Physics‑only:** same embedding as the base, but pretraining loss includes physical constraints.
> - **Group‑only:** uses our grouped spectral embedding with SimMIM on the spatial dimension.
> - **Refined‑MixMAE‑only:** uses our refined MixMAE pretraining with native embedding and no physics constraints in the loss.
> - **With‑all:** combines refined MixMAE, physics priors, and grouped embedding.
> - We also included a **non‑trained SwinV2 (scratch)** for comparison.
>
> **Takeaways.**
> - Physics‑only improves over the SimMIM base consistently across all tasks: **+2.92 F1** (OSCD), **+2.07 mIoU** (SegMunich), **+1.22 F1** (EuroSAT), confirming that physics‑informed objectives contribute independently.
> - **With‑all** maintains gains over the base while enabling the **efficiency benefits** of grouped embedding and refined MixMAE.
> - **Group‑only** and **refined‑MixMAE‑only** do not explain the observed accuracy gains, underscoring that the **physics priors are a major driver** of performance, while other components mainly deliver efficiency.
>
> We will add a short note in §4.4 to clearly attribute accuracy gains to the physics‑informed losses and efficiency to the architectural choices.
>
> ## Point 2: Physics‑informed losses on MAE SOTA backbones
> Thank you for this suggestion. To test generality beyond PhySwin, we added our two physics losses to a ViT-B model pretrained with standard MAE (no grouped embedding, no refined MixMAE).
>
> | Backbone                | OSCD F1 %          | SegMunich mIoU %    | EuroSAT F1 %        |
> |------------------------|--------------------:|---------------------:|---------------------:|
> | ViT-B MAE                | 50.46 ± 0.505       | 42.00 ± 0.840        | 94.78 ± 1.896        |
> | ViT-B MAE + Physics  | **52.41 ± 0.524**   | **44.17 ± 0.883**   | **95.20 ± 0.952**    |
>
> We can observe that the gains arise again. This is align with our statements in the paper that the physics regularizers encourage inter‑band consistency and suppress out‑of‑range reconstructions, guiding features toward a physically plausible direction and faster adaptation to valid ranges. We will add a brief note in §4.4 as a baseline for refined MixMAE and Spectral Group Masking.
>
> *(Compute note: these runs follow the same reduced‑epoch style, 15 epochs, used in our ablations; we did not retune hyperparameters for the ViT + physics.)*
>
> ## Point 3: Clarifying importance, physics link and architectural emphasis
>
> Thank you for this constructive critique. We will rebalance the presentation to make the motivation and contributions clearer.
>
> **Why it matters**
> We observe a growing number of foundation models emerging in EO, with a strong focus on scaling parameters and pretraining datasets to push performance. While this trend has yielded impressive accuracy, there is also a rapidly growing interest in **deployment‑level ML for EO**. This interest is motivated by the fact that **large amounts of EO data can be discarded or delayed because of the high cost and latency of communicating with ground stations**. Our work is designed with this balance in mind. We aim not only for strong accuracy but also for efficiency at both training/finetuning and inference. Specifically, we introduce a **grouped spectral embedding** to reduce token counts without sacrificing spectral structure and refine **MixMAE** to maintain Swin’s window‑shifted context while being more efficient during the pretraining. Together with our physics‑guided priors, these components address the dual challenges of learning meaningful multispectral representations and achieving the throughput/memory profile required for practical deployment.
>
>
> **Physics priors are not generic "smoothing" or "clipping"**
> - **Spectral smoothness** encodes the well‑established observation that surface reflectance varies smoothly with wavelength; we implement this as a first‑order difference on *adjacent bands in spectral order* and apply it to the reconstructed reflectance during pretraining.
> - **Energy bounds** encode the physical boundedness of reflectance, serving as a soft barrier that keeps reconstructions in a realistic range while tolerating sensor noise.
>
> These losses operate on **reconstructed reflectance vectors**, tying them directly to spectral physics rather than generic pixel heuristics. Our ablations (Point 1) and MAE‑backbone test (Point 2) show consistent gains **even without architectural changes**, indicating the priors improve representation quality in a plug‑in manner.
>
> **Architectural choices**
> - **Backbone:** SwinV2 for hierarchical, windowed attention → near‑linear scaling and strong dense‑task transfer.
> - **Refined MixMAE:** we retain **SW‑MSA** and coordinate masking with window shifts, preserving cross‑window context while avoiding [MASK] tokens.
> - **Token‑efficient grouped spectral embedding (+MaskSpec):** preserves rich band structure **without increasing token count**, delivering high throughput with robust features.
>
> Our efficiency results in the paper highlight that these choices approach ConvNet‑level throughput while remaining competitive on accuracy.
>
> We will (i) move a concise design‑rationale paragraph to the end of §3, explicitly mapping each component to accuracy vs efficiency goals; (ii) add a short physics‑motivation paragraph in §3.1 linking the losses to spectral smoothness and bounded reflectance. We hope these changes make the importance and novelty clearer.
>
> ## Point 4: Additional suggestions and clarity improvements
>
> We thank the reviewer for these practical suggestions and will address each point:
>
> 1. **Marking replicated results:** We will clearly label in all tables which results are independently reproduced by us and which are reported directly from the original papers.
> 2. **Alternative masking strategies:** While orthogonal to our current design, we agree that exploring single‑component masking (e.g., uniformly sampled masks across spectral groups) could offer additional insights into feature robustness and inter‑group relationships. We will explicitly note this as an interesting direction for future work.
> 3. **Figure 1 captions:** We will expand the captions for Figure 1a–c to better explain:
>    - **Figure 1a:** The bounded reflectance distribution in Sentinel‑2 multispectral data and how this motivates the **energy‑bound loss**.
>    - **Figure 1b:** The smooth transitions across adjacent MS bands, showing empirical spectral differences and motivating the **spectral‑smoothness loss**.
>    - **Figure 1c:** The trade‑off between accuracy and inference throughput for EO foundation models, positioning PhySwin’s improvements.
>
>    These expanded captions will explicitly link the visuals to the physics‑informed objectives (§3.1) and the overall efficiency‑accuracy motivation of PhySwin.

---

> > ### Comment · Reviewer_FjVd · 2025-08-04
> >
> > I remain unconvinced by the motivation behind the Physics priors. However, the authors have successfully addressed all of my concerns.
> >
> > I will raise my rating accordingly.

---

> > > ### Author Response · Authors · 2025-08-04
> > > **Thank you**
> > >
> > > We sincerely thank Reviewer FjVd for taking the time to engage with our rebuttal and for acknowledging that we have successfully addressed the concerns raised. We appreciate your updated rating and your engagement throughout the review process.

---

### Official Review · Reviewer_HaH9 · 2025-06-27

**Clarity:** 4
**Significance:** 3
**Originality:** 3
**Rating:** 5
**Confidence:** 4

**Summary:**

The paper proposes a new foundation model for multispectral remote sensing data which introduces three modifications with respect to previous work in its architecture/pre-training process: (1) it integrates radiometric physical constraints into the pre-training loss (bounding reflectance values to be on fixed range due to energy conservation), (2) it uses an efficient modified MixMAE scheme, which coordinates the masking patterns with the shifted windows in SWIN, (3) a spectral embedding technique that groups spectral bands and uses a lightweight embedding function per group that are then concatenated to create a single embedding vector for each spatial location, maintaining the final token count fixed, independent of the number of channels.
They compare this method with other foundation models for multispectral remote sensing data on four different types of downstream tasks: semantic segmentation (2 benchmark datasets), change detection (2 benchmark datasets), scene classification (2 benchmark datasets) and multi-label land cover classification (1 benchmark dataset), both in terms of accuracy metrics (appropriate for each task) as well as inference efficiency. They perform ablation studies to verify the effect of each of the modifications proposed also both in terms of accuracy metrics and on inference efficiency.

**Questions:**

1. Why have you ignored the temporal aspect of remote sensing images in pre-training? Do you think it is not important to model this? Would it be possible to modify your model to include multiple images as input?
2. How have you selected the models to compare against? There are many other models (see link above) which have not been included such as DOFA, Prithvi-v2, DINO, AnySat.
3. Thinking about a “physically-grounded” model for remote sensing, is bounding reflectance values to be on fixed range due to energy conservation the only physical constraint possible?
4. Are there ways to trade-off accuracy versus efficiency in your model? What is the effect of the sizes of the network, number of parameters?

**Ethical Concerns:**

["NO or VERY MINOR ethics concerns only"]

**Final Justification:**

The author's response has satisfactorily addressed the negative points I had raised. With the new results that they presented both in terms of the role of the "physics-informedness" of the model and comparisons with other SOTA models, I believe this is a technically solid paper with significant result for the area of foundation models for earth observation.

**Limitations:**

Yes

**Paper Formatting Concerns:**

I did not notice any major formatting issues in this paper.

**Quality:**

3

**Strengths And Weaknesses:**

Strengths:
1. The paper is well-written and describes very clearly what are the proposed improvements relatively to previous work and the rationale behind the changes.
2. The proposed modifications seem technically sound.
3. The experimental analysis and ablation studies are well designed.
4. The accuracy results for semantic segmentation and change detection show that the model is on par or better than the other FMs compared.
5. The efficiency results in terms of throughput in inference are very compelling.

Weaknesses:
1. The proposed changes are incremental and heavily build upon previous work.
2. The claim of being “physically-grounded” is a little strong given that they are only introducing a bound on the reflectance value in the training loss).
3. For the scene classification and multi-label land cover classification task, they say that the results are competitive “near-SOTA” in terms of accuracy, but in fact the differences are quite large in terms of percentage points (for BigEarthNet it is 4 percentage points), so it might not be worth having a faster model which is actually significant worse.
4. There are many recent foundation models for remote sensing data (see https://github.com/Jack-bo1220/Awesome-Remote-Sensing-Foundation-Models) and they only compare against a few, leaving out ones which seem to have better results in terms of accuracy (DOFA, Prithvi-v2, DINO, AnySat,…), although it is difficult to compare because there are no standard benchmarks for this type of model (GeoBENCH is an attempt to establish that).
5. Although there is a temporal aspect to remote sensing images, this paper completely ignores this, where most of the recent models take that aspect into account.

---

> ### Author Rebuttal · Authors · 2025-07-30
>
> ## Point 1: "Physically‑grounded" claim clarification
>
> Thank you for raising this. We would like to clarify that our physical component comprises **two priors**, not only a bound:
>
> - **Spectral smoothness:** enforces continuity across **adjacent spectral bands**, reflecting the well‑documented smooth variation of surface reflectance with wavelength (Fig. 1b, §3.1).
> - **Energy bounds:** applies a **soft reflectance‑range prior** so reconstructions remain physically plausible (Fig. 1a, §3.1).
>
> As suggested by reviewer **FjVd**, we also ran the new ablations as the evidence that **the priors matter (beyond architecture)**. More information can be found in our answers of reviewer **FjVd**.
>
> **A) Isolating the physics priors within our setup** (PhySwin‑T, 15‑epoch for all variants due to compute limits). *(As suggested by reviewer **qjEV**, we report mean ± std over 3 seeds 42/177/892)*
>
> | Variant               | OSCD F1 %           | SegMunich mIoU %     | EuroSAT F1 %        |
> |----------------------|---------------------:|----------------------:|---------------------:|
> | Base (SimMIM)        | 49.31 ± 0.986        | 42.64 ± 0.853         | 92.83 ± 0.371        |
> | **Physics only**     | ***52.23 ± 0.261***    | ***44.71 ± 0.894***     | ***94.05 ± 0.940***    |
> | Group only           | 48.50 ± 0.485        | 40.22 ± 0.804         | 91.47 ± 1.371        |
> | Refined‑MixMAE only  | 49.06 ± 0.981        | 43.10 ± 0.862         | 92.61 ± 1.852        |
> | With ALL             | **52.07 ± 0.521**        | **43.29 ± 0.433**         | **93.73 ± 0.937**        |
> | Base (Scratch)       | 44.90 ± 0.449        | 40.37 ± 0.323         | 89.24 ± 1.339        |
>
> This isolates the contribution of the priors: they improve accuracy **without** grouped embedding or refined MixMAE.
>
> **B) Adding the priors to a separate backbone** (ViT‑B + MAE, 15 epoch).
>
> | Backbone               | OSCD F1 %          | SegMunich mIoU %     | EuroSAT F1 %        |
> |-----------------------|--------------------:|----------------------:|---------------------:|
> | ViT MAE               | 50.46 ± 0.505       | 42.00 ± 0.840         | 94.78 ± 1.896        |
> | **ViT MAE + Physics** | **52.41 ± 0.524**   | **44.17 ± 0.883**     | **95.20 ± 0.952**    |
>
> These arise that the physics regularizers encourage **inter‑band consistency** and **suppress out‑of‑range reconstructions**, guiding features toward a **physically plausible direction** and faster adaptation to valid value ranges.
>
> Furthermore, to avoid overstatement, we will adjust the wording to **“physics‑informed priors”** and add explicit pointers to **Fig. 1a–b** and **§3.1** where these losses are defined and motivated.
>
> ## Point 2: "Incremental changes"
>
> We appreciate the concern and will make the **design rationale** more explicit. Our contributions target the dual goal of **accuracy + deployment‑level efficiency** for multispectral EO, and they work **together** rather than as small tweaks in isolation:
>
> - **Token‑efficient grouped spectral embedding.**
>   Unlike standard patch embeddings whose token count grows with channels, our grouped spectral embedding **keeps the token count fixed** while preserving band structure via lightweight per‑group mappings and concatenation. This directly reduces memory/FLOPs without discarding spectral information. Meanwhile the randomly masking spectral groups during the pretraining ecourage the model to learn from different groups, avoiding the bias in features.
>
> - **Refined MixMAE that preserves SW‑MSA.**
>   We coordinate MixMAE’s masking with Swin’s shifted windows, **retaining cross‑window context** and **eliminating [MASK] tokens** that otherwise slow hierarchical transformers. This lets us **pretrain Swin‑v2 faithfully** and then fine‑tune without modifying the architecture.
>
> - **Physics‑informed priors (two losses).**
>   Beyond architecture, we introduce **spectral smoothness** (adjacent‑band continuity) and **energy bounds** (plausible reflectance range). These priors **improve accuracy independently** of architectural changes (see Point 1 ablations and the **ViT‑MAE + Physics** result).
>
> **Action we will take:** we will refine **design‑rationale paragraph** for mapping each component to its role (grouped embedding → token/compute efficiency; refined MixMAE → Swin‑compatible pretraining with high throughput; physics priors → accuracy/robustness), and we will put these connections earlier to avoid the impression of small, incremental tweaks. Thank you for raising this point.
>
> ## Point 3: Classification results and "near‑SOTA" concern
>
> Thank you for flagging this. The cited **+4 pts** comes from **literature‑reported** Skysense numbers; **Skysense is not open‑sourced**, so we cannot evaluate it under the **same** protocol/settings.
>
> To ensure **scope and fairness**, and **as suggested by reviewer HaH9 and HFfE**, we moved classification to the open‑sourced **PANGAEA/GeoBench** framework and now report **macro‑F1** in addition to accuracy/mAP. Under this unified setup, we get **TABLE 1** in our response for reviewer **qjEV**.
>
> These standardized, reproducible runs show that **PhySwin‑B is still top‑tier** compared with the most recent baselines while maintaining strong efficiency. We believe this can address the "near‑SOTA" concern under a unified evaluation protocol.
>
> **Our actions:** We will label **ours (reproduced)** vs **from literature** in all tables for transparency. Some strong baselines (e.g., non‑open‑sourced checkpoints) may reflect recipe differences.
>
> ## Point 4: Baseline coverage concern
> Thanks for pointing this out. To address it fairly and reproducibly, we moved our evaluation to the **PANGAEA/GeoBench** framework and added recent EO FMs to the comparison, following the unified recipe.
>
> **TABLE 1: PANGAEA results (S2 benchmarks).**
>
> | Model | MADOS | CROPMAP | PASTIS | SEN1FLOODS11 |
> |---|---:|---:|---:|---:|
> | CROMA | 57.81 ± 0.34 | 44.17 ± 0.78 | ***15.83 ± 0.17*** | 87.12 ± 1.18 |
> | DOFA | ***64.40 ± 0.19*** | 38.48 ± 0.53 | 12.58 ± 0.19 | 85.42 ± 0.83 |
> | PRITHVI | 41.66 ± 0.70 | **50.60 ± 0.21** | 11.61 ± 0.23 | **87.38 ± 0.95** |
> | DINO | 55.05 ± 0.82 | 47.93 ± 0.52 | 15.40 ± 0.18 | 86.11 ± 0.51 |
> | SatLasNet | 52.80 ± 0.68 | 45.04 ± 0.36 | 13.29 ± 0.21 | 82.96 ± 1.57 |
> | **PhySwin‑T** | 55.75 ± 0.50 | 44.37 ± 0.79 | 14.31 ± 0.14 | 85.93 ± 0.77 |
> | **PhySwin‑B** | **63.06 ± 0.88** | ***51.83 ± 0.31*** | **15.69 ± 0.25** | ***88.30 ± 0.41*** |
>
> **Takeaway.** Under a standardized pipeline, **PhySwin‑B is Top‑1/Top‑2 across all four tasks**, while preserving efficiency.
>
> **Reproducibility.** We use PANGAEA defaults **with S2 bands** and hyperparameter settings; runs can be reproduced with the following command:
>
> ```bash
> torchrun --nnodes=1 --nproc_per_node=1 pangaea/run.py \
>   --config-name=train \
>   decoder=seg_upernet preprocessing=seg_resize criterion=cross_entropy \
>   task=segmentation task.trainer.n_epochs=80 batch_size=16 \
>   encoder=ENCODER dataset=DATASET
>   ```
>
> Note: Some models (e.g., terramind_large) require manual fixes to download and run experiments, so we skip them for now. We are continuously adding those that can be executed in PANGAEA and will include any newly completed entries in the later discussion and revised version if required. More experimental settings can be found in our answers for reviewer **HFfE**.
>
> ## Point 5: Temporal aspect
>
> Thank you for raising this. We totally agree that **Temporal modeling is important** for EO (e.g., crop phenology, flooding dynamics).
>
> **Why we did not include it in this release.**
> Our first objective was to balance **accuracy and deployment‑level efficiency** under a fixed token/compute budget (on‑satellite/edge constraints). Multi‑temporal pretraining multiplies the token count and memory footprint; we therefore prioritized an **efficient multispectral encoder** and validated it thoroughly under a standardized pipeline. Note that while pretraining is single‑timestamp, our **change‑detection downstream head** already consumes **bi‑temporal inputs** in fine‑tuning/evaluation following the benchmark recipe. Meanwhile, this performance ranks among the top in all SOTA benchmark tests.
>
> For question **Is it important / can the model handle multiple images?** The answer is Yes. PhySwin is **readily extensible to multi‑temporal inputs**, and this is **under active development**. Concretely, we are trying to implement (i) **temporal patching/merging** to keep the token budget roughly fixed, and (ii) **spatio‑temporal masking** aligned with SW‑MSA. These are directly compatible with our PhySwin design.
>
> We will explicitly state in the paper that this version focuses on **efficient pretraining**, and we will list **temporal extensions** (temporal patching/merging + spatio‑temporal masking) as **future work**. This roadmap preserves the efficiency goals while adding the temporal dimension.
>
> ## Point 6: Accuracy–efficiency trade‑offs
>
> Yes, PhySwin is able to trade accuracy for speed/memory.
>
> **Practical aspects to tune the trade‑off.**
> - **Model size:** choose **T** for edge/real‑time or **B** for higher accuracy with moderate extra cost.
> - **Token budget:** adjust **spectral grouping granularity** (fewer groups → fewer params/tokens; more groups → richer per‑band specialization).
> - **Swin window/shift settings:** larger windows capture more context at extra compute.
>
> For example, across our new PANGAEA/GeoBench results (Tables 1–2), **PhySwin‑B** consistently outperforms **PhySwin‑T** on segmentation, change detection, and classification, while **PhySwin‑T** delivers higher throughput and lower memory (Table 4 in the paper). This shows **scaling model size helps accuracy**, but we intentionally keep both variants **compact** to meet our deployment constraints.
>
> We will add a brief paragraph in the paper summarizing these tunes' effects to help readers select a configuration aligned with their compute budget.

---

> ### Author Response · Authors · 2025-08-05
> **Follow-Up**
>
> We thank the reviewer for the valuable comments. We would appreciate it if you could let us know whether our response sufficiently addresses the concerns.

---

> ### Author Response · Authors · 2025-08-09
>
> We thank the reviewer for the valuable comments. We would appreciate it if you could let us know whether our response sufficiently addresses the concerns.

---

### Official Review · Reviewer_qjEV · 2025-07-02

**Clarity:** 4
**Significance:** 4
**Originality:** 4
**Rating:** 5
**Confidence:** 5

**Summary:**

This paper introduces PhySwin, a foundation model developed for multispectral (MS) Earth observation (EO) imagery that addresses key limitations in current remote sensing foundation models (RSFMs), namely computational inefficiency and the neglect of physical/knowledge priors. The model incorporates three core innovations: physics-informed pretraining, which adds spectral smoothness and energy conservation constraints as regularizations; a refined MixMAE method adapted for the hierarchical SwinV2, utilizing shifted window attention to improve scalability and efficiency; and a token-efficient spectral embedding strategy that groups similar spectral bands and applies spectral group masking (MaskSpec) to retain spectral information without increasing token count. PhySwin is trained on +1M Sentinel-2 images, and it shows improved performance on key EO tasks such as semantic segmentation and change detection with computational complexity.

**Questions:**

- Spectral smoothness is a novel way of using physical information in the model training. However, misregistration of various spectral bands can punish $L_{smooth}$. How can this be addressed? Is this something that we need to accept as a limitation?

- L141-142: It’s unclear how the claim “tokens attend only to others from the same source image, as dictated by M” is enforced. Equation (3) governs the reconstruction loss, but it does not prevent cross-source attention during encoding. Did you explicitly create an attention mask using M to restrict token interaction within each source? If so, I suggest adding 1–2 sentences in the text to clarify how this constraint is implemented in the encoder. In addition, the dual reconstruction loss in Eq(3) allows the model to learn to reconstruct missing information from real images, it would be good to add this.

- Why Table 4 is missing SkySense, despite SkySense showing high performance in some tasks (Tables 2, and 3)?

- [Minor] Figure 3 is missing a color legend for labels, please add.

**Ethical Concerns:**

["NO or VERY MINOR ethics concerns only"]

**Final Justification:**

The authors have successfully addressed my questions, and provided new results that addresses the weaknesses I raised.

**Limitations:**

Yes.

**Paper Formatting Concerns:**

No.

**Quality:**

4

**Strengths And Weaknesses:**

The proposed model, PhySwin, is a novel advancement that integrates physically grounded learning in the training stage, incorporating radiometric constraints (bounded reflectance and spectral smoothness) to enhance generalization and interpretability. A masking strategy that combines spatial and spectral masking further strengthens feature resilience and richness. Its efficiency-focused architecture, built on SwinV2 and a refined MixMAE framework, delivers significant computational advantages. The model demonstrates enhanced performance across multiple Earth observation tasks, with gains such as +1.32% in segmentation mIoU and +0.80% in change detection F1 score. These benefits are validated through extensive experiments on diverse benchmarks.
PhySwin some limitations too as it underperforms on all multi-label and scene classification tasks when compared to ViT-based baselines. Additionally, the study does not explore scaling up the model, leaving open the question of how larger PhySwin variants might perform on more complex or diverse tasks (while authors list this as a limitation). Finally, the paper does not report uncertainty estimates or statistical significance tests, making it difficult to evaluate the variability or robustness of the reported results.

---

> ### Author Rebuttal · Authors · 2025-07-30
>
> ## Point 1: Classification performance clarification
>
> Thank you for raising this. The cited relatively large **+4 pts** gap comes from **literature‑reported** Skysense numbers. **Skysense is not open‑sourced**, so we cannot evaluate it under the same protocol. Following other reviewers’ suggestions, we now use the **unified, open‑sourced PANGAEA/GeoBench** framework and report **macro‑F1** (in addition to Acc/mAP), with **mean ± std over 3 seeds (42/177/892)**.
>
> **TABLE 1: Classification under P/GEO (EuroSAT & BigEarthNet‑subset).**
>
> | Model | **EuroSAT** Acc | **EuroSAT** macro‑F1 | **BigEarthNet (subset)** mAP | **BigEarthNet (subset)** macro‑F1 |
> |---|---:|---:|---:|---:|
> | CROMA | 93.13 ± 0.18 | 93.13 ± 0.18 | ***72.50 ± 0.73*** | ***64.90 ± 0.65*** |
> | DOFA | 94.85 ± 0.28 | 94.85 ± 0.28 | 69.50 ± 0.39 | 62.40 ± 0.25 |
> | PRITHVI | 92.66 ± 0.85 | 92.66 ± 0.85 | 64.50 ± 0.65 | 58.30 ± 0.58 |
> | DINO | 93.41 ± 1.06 | 93.41 ± 1.06 | 68.40 ± 0.37 | 60.30 ± 0.60 |
> | SatLasNet | **97.81 ± 0.95** | **97.81 ± 0.95** | 70.00 ± 0.70 | **64.80 ± 0.30** |
> | PhySwin‑T | 96.32 ± 0.16 | 96.32 ± 0.16 | 68.20 ± 0.68 | 61.90 ± 0.24 |
> | **PhySwin‑B** | ***98.17 ± 0.19*** | ***98.17 ± 0.19*** | **71.60 ± 0.43** | 64.10 ± 0.27 |
>
> *Notes:* (i) GeoBench uses a **subset** of BigEarthNet (20k train / 1k test), so numbers differ from full‑dataset training; (ii) **Provenance** will be labeled in all tables (**ours (reproduced)** vs **from literature**).
>
> As shown, **PhySwin‑B** remains **top‑tier** and aligns with our paper’s comparisons (excluding Skysense), while maintaining the **efficiency** needed for deployment‑level use. Similar discussions can be also seen in point 3 for reviewer **HaH9**.
>
> ## Point 2: Uncertainty estimates & significance
>
> Thank you for noting this. In our updated experiments we now **report mean ± std over 3 random seeds (42, 177, 892)** for all tasks and tables (including newly added experiments based on PANGAEA/GeoBench). Across seeds, results are **stable**, normally less than 1% (small std), and we will **add seed info and mean ± std** to table captions in the paper and supplement. Additional results can also be found in the tables included in our answers for other reviewers.
>
> These additions address the reviewer’s request for uncertainty reporting and statistical support.
>
> ## Point 3: Spectral smoothness vs. band misregistration
>
> Thank you for the thoughtful question. We **agree misregistration can bias a spectral‑smoothness prior** if adjacent bands are not perfectly aligned.
>
> Currently, in our pipeline, the smoothness loss is implemented as a **soft penalty** (with moderate weight) applied to the reconstructed reflectance vector per pixel. This means it serves as a gentle prior rather than a strict constraint. However, we acknowledge that residual misregistration remains a limitation of the current formulation. **As future work**, we plan to explore **misregistration‑robust smoothness variants**, or alternatively, more straightforward approaches such as data preprocessing or filtering methods to avoid this happens.
>
> ## Point 4: Cross‑source attention constraint (L141–142) & dual reconstruction
>
> Thank you for the question. We will clarify how we ensure "tokens attend only to others from the same source image, as dictated by **M**."
>
> During pretraining with mixing, we build an **attention mask** from the mixing map **M** and apply it to **every self‑attention layer** (both W‑MSA and SW‑MSA). Concretely, if tokens *i* and *j* come from different source images according to **M**, the attention score entry is set to **the very large nagative value** before softmax (i.e., a **block‑diagonal attention mask** over sources). In addition, window partitioning packs tokens **per source** so that windows never span tokens from two sources; the mask remains active after the shift in SW‑MSA to prevent any cross‑source leakage.
>
> For **Dual reconstruction (Eq. 3)**, we will add one sentence noting that the **dual reconstruction loss** reconstructs the masked regions of **each source image from its own unmasked context**, i.e., the model is trained to recover missing content from *real within‑source evidence*, not from cross‑source tokens.
>
> We will include more details in Section 3 and new pseudo‑code in the Supplementary.
>
> ## Point 5: Why Skysense is absent from Table 4
> As noted in **Point 1**, **Skysense is not open‑sourced**, so we cannot reproduce it under our unified pipeline. **Table 4** includes only models we can run end‑to‑end.
>
> ## Point 6: Figure 3 color legend
> Thanks for catching this. We will **add a color legend** to Figure 3 and clarify the caption. Quick details:
> - **Top:** OSCD change‑detection masks (foreground/background).
> - **Bottom:** **SegMunich** (13 LULC classes) and **Dyna.-S2** semantic segmentation—colors will follow each dataset’s **official palette** and be listed in the legend.
>
> We will update the caption accordingly.

---

> ### Author Response · Authors · 2025-08-05
> **Follow-Up**
>
> We thank the reviewer for providing the final score and for the valuable comments on our work. We would appreciate it if you could let us know whether our response sufficiently addresses your concern.

---

> > ### Comment · Reviewer_qjEV · 2025-08-05
> >
> > Yes, all sounds good to me, and I have finalized my score accordingly.

---

### Note · Authors · 2025-08-12

We thank the reviewers and AC for the constructive feedback and productive discussion. We are pleased that all major concerns were addressed during the rebuttal, with multiple reviewers confirming their points were resolved and updating their ratings accordingly.

Key points clarified in the rebuttal and discussion:
- Ensured fair and up-to-date comparisons against the latest baselines using the standard open-source benchmark framework.
- Verified novelty and clear positioning against prior RSFMs.
- Confirmed that performance improvements are consistent across benchmarks and not due to confounding factors.
- Clarified ablation study methodology and statistical reliability.
- Discussed limitations and future directions clearly.

Our work delivers state-of-the-art accuracy with substantial efficiency gains on diverse EO tasks, using public datasets and a reproducible pipeline. We will release code, models, and data preprocessing scripts to facilitate further research and adoption.

We appreciate the reviewers’ engagement, which strengthened the paper, and we believe the final version makes a clear and valuable contribution to the NeurIPS community.

---

### Decision · Program_Chairs · 2025-09-17

**Decision:**

Accept (poster)

**Comment:**

This paper presents PhySwin, a foundation model for multispectral data that integrates physical priors and demonstrates computational efficiency.

The work is commended for its strategic masking strategy as well as effective integration of physical learning in the training stage.

Weaknesses included desire for further details on experimental results, which were provided in the rebuttal period. Some other masking strategies suggested remain to be tested.

Overall, all reviewers agreed this is a strong contribution and carefully discussed the rebuttal with authors with appropriate technical detail, upon which all major concerns were addressed.